

# Measurement and modeling of the multi-wavelength optical properties of uncoated flame-generated soot

Sara D. Forestieri,[1, #] Taylor M. Helgestad[1,#] Andrew Lambe,[2,3] Lindsay Renbaum-Wolff[2], Daniel A. Lack,[4,5,^] Paola Massoli,[2] Eben S. Cross,[6,&] Manvendra K. Dubey,[7] Claudio Mazzoleni,[8] Jason Olfert,[9] Andrew Freedman,[2] Paul Davidovits,[3] Timothy B. Onasch,[2,3] Christopher D. Cappa[1]

[1]Department of Civil and Environmental Engineering, University of California, Davis, CA 95616
[2]Aerodyne Research Inc., Billerica, Massachusetts, USA, 01821
[3]Chemistry Department, Boston College, Boston, MA, USA, 02467
[4]NOAA Earth System Research Laboratory, Boulder, CO, USA, 80305
[5]University of Colorado, Cooperative Institute for Research of the Environmental Sciences, Boulder, CO, USA, 80305
[6]Department of Civil and Environmental Engineering, Massachusetts Institute of Technology, Cambridge, Massachusetts, USA
[7]Los Alamos National Laboratory, Los Alamos, NM, USA
[8]Deparment of Physics and Atmospheric Sciences Program, Michigan Technological University, Houghton, MI, USA
[9]Department of Mechanical Engineering, University of Alberta, Edmonton, Alberta, Canada
[#] Now at: California Air Resources Board, Sacramento, CA, USA
[^] Now at: Transport Emissions, Air Quality and Climate Consulting, Brisbane, Australia
[&] Now at: Aerodyne Research Inc., Billerica, Massachusetts, USA, 01821

Correspondence to: Sara Forestieri (*sara.forestieri@arb.ca.gov*) or Christopher Cappa (cdcappa@ucdavis.edu)

## 1 Abstract

Optical properties of flame-generated black carbon (BC) containing soot particles were quantified at multiple wavelengths for particles produced using two different flames, a methane diffusion flame and an ethylene premixed flame. Measurements were made for: (i) nascent soot particles, (ii) thermally denuded nascent particles, and (iii) particles that were coated then thermally denuded, leading to collapse of the initially lacy, fractal-like morphology. The measured mass absorption coefficients (MAC) depended on soot maturity and generation, but were similar between flames for similar conditions. For mature soot, here corresponding to particles with volume-equivalent diameters > ~160 nm, the MAC and absorption Angstrom exponent values



were independent of particle collapse while the single scatter albedo increased. The MAC values for these larger particles were also size-independent. Effective, theory-specific complex refractive index (*RI*) values are derived from the observations with two widely-used methods: Lorenz-Mie theory and the Rayleigh-Debye-Gans (RDG) approximation. Mie theory systematically under-

predicts the observed absorption cross-sections at all wavelengths for larger particles (with $x > 0.9$) independent of the complex *RI* used, while RDG provides good agreement. Importantly, this implies that the use of Mie theory within air quality and climate models, as is common, likely leads to under-predictions in the absorption by BC, with the extent of under-prediction depending on the assumed BC size distribution and complex *RI* used. We suggest that it is more appropriate to

assume a constant, size-independent (but wavelength-specific) *MAC* to represent absorption by uncoated BC particles within models.

## 2    Introduction

Soot particles, which contain light-absorbing black carbon (BC), are a byproduct of incomplete combustion of fossil fuels and biomass. These particles affect climate directly by absorbing and

scattering solar radiation (Bond et al., 2013) and indirectly by acting as cloud condensation nuclei, especially following chemical processing (Lohmann and Feichter, 2005). Soot particles absorb shortwave radiation and have an overall warming effect on climate. Although the exact magnitude of the climate impacts of BC remain uncertain, one estimate puts top-of-the-atmosphere direct forcing by BC as high as 0.9 W m$^{-2}$, which is comparable in magnitude to that of $CO_2$ (Ramanathan

and Carmichael, 2008).

One challenge in modelling the optical properties of soot, and of the BC-component in particular, derives from BC having a complex, fractal-like structure, being an agglomerate of small "spherules" (Medalia and Heckman, 1969). One theory that is commonly used in climate models to calculate BC optical properties is Lorenz-Mie Theory (hereafter, Mie theory), which makes the

physically unrealistic assumption that soot particles are spherical (Bohren, 1983). This theory is widely used in climate models (Bond et al., 2013) in part because it does not require any details about the number of spherules or the arrangement of the spherules within the agglomerate, but also because it is compatible with calculations for other spherical aerosol types. A variation on Mie theory, the Rayleigh-Debye-Gans (RDG) approximation, is also often used to model the optical




properties of BC (Sorensen, 2001), albeit not by climate models. In RDG, the agglomerate absorption cross section ($\sigma_{abs}$) is the product of $\sigma_{abs}$ for individual spherules and the number of spherules in the agglomerate. As such, RDG neglects spherule-to-spherule interactions and the mass absorption coefficient (*MAC*) of an individual spherule is equal to that of the overall particle.

(The *MAC* is the absorption cross-section normalized by the particle mass.) There are more complex methods for calculating soot particle optical properties, including the T-matrix method (Mackowski and Mishchenko, 1996) and the discrete dipole approximation (DDA) (Purcell and Pennypacker, 1973), which account for interactions between spherules. Given that these more advanced methods require detailed information on the shape of the soot particles and are

computationally intensive, they are not practical for climate models. The derived effective refractive indices used as inputs for these models are theory-specific and it is necessary to have experimentally constrained effective refractive indices for both RDG and Mie theory if they are to be employed in climate models. For example, Bond et al. (2006) suggested that BC can be described using an $MAC = 7.5$ m$^2$ g$^{-1}$ at 550 nm and a complex $RI = 1.95 - 0.79i$. However, as they

show, the maximum *MAC* calculated from Mie theory using this refractive index is only 7.2 m$^2$ g$^{-1}$ over a very narrow range of particle sizes and is much smaller in general, with a value of 4.9 m$^2$ g$^{-1}$ in the small particle limit, where RDG applies (assuming $\rho = 1.8$ g cm$^{-3}$). In other words, there can be an inconsistency between the oft used *MAC* and complex *RI*. This illustrates the need for theory-specific effective refractive index values and a fuller exploration of the robustness of

commonly used optical models.

Our work investigates the ability of two optical models, Mie theory and the RDG approximation, to reproduce observed soot optical properties for particles composed primarily of BC. The observations include light absorption and extinction coefficients of soot particles produced from methane diffusion and ethylene premixed flames, measured over four different studies at multiple

wavelengths. The impact of shape on soot particle optical properties is also examined. The soot particles sampled during these studies serve as a proxy for different types of soot particles in the ambient atmosphere. Recommended theory-specific complex RI values for BC-dominated soot particles are provided. However, we ultimately suggest that atmospheric models should consider adopting observationally constrained, wavelength-specific constant *MAC* values for BC rather

than calculating the optical properties from optical theories.



## 3    Experimental: The Black Carbon Studies

The measurements reported here were made during a series of laboratory intensive studies that took place at Boston College (BC2, BC3 and BC4) in 2008, 2012 and 2015, respectively, and Aerodyne Research (BC3+) in 2014. Below, we provide details of soot particle generation and the measurements made. An experimental schematic is provided in Figure S1.

### 3.1    Soot particle generation

Soot particles were produced using two different flame sources and fuel types. During BC3, BC3+, and BC4, most experiments were conducted using particles produced from an inverted co-flow diffusion flame operating on methane with a sheath flow mixture of $O_2$ and $N_2$ (Stipe et al., 2005) with a fuel equivalence ratio, $\phi$, = 0.7. These are referred to as the "methane diffusion flame" experiments, and have been combined into a single dataset since the sampling and generation were similar in all.

During BC2 and for a small number of experiments during BC3+, particles were produced using a McKenna flat-flame burner from combusion of premixed $C_2H_4$ (ethylene), $O_2$ and $N_2$ with $\phi$ = $2.0 \pm 0.2$. These are referred to as the "ethylene premixed flame" experiments. Particles were sampled at a nominal height of 5 cm above the burner during BC3+, but at a nominal height of 20 cm above the burner during BC2 (Cross et al., 2010).  As such, the results from the two ethylene premixed flame have been kept separate because particles were sampled from the flame differently in the two studies.

The soot particles produced from these two flames exhibited different properties. For example, the organic (OC) mass fraction of the nascent (i.e. freshly emitted and unprocessed) ethylene premixed flame soot particles in BC2 was ~0.26 (Cross et al., 2010), whereas the OC fraction of nascent methane diffusion flame particles was < 0.01. Consequently, upon heating to >200°C the per-particle mass of the ethylene premixed flame soot particles decreased while the methane diffusion flame soot particles were unaffected.

The sampling and/or burner conditions were modified during these studies to generate monodisperse soot particles volume equivalent diameter ($d_{p,VED}$) less than 160 nm. The $d_{p,VED}$ is





the diameter calculated from the per-particle mass assuming that particles have spherical morphology and the material density of the particles is 1.8 g/cm$^3$ (Mullins and Williams, 1987; Wu et al., 1997; Bond and Bergstrom, 2006):

$$d_{p,VED} = \left(\frac{6m_p}{1.8 \cdot \pi}\right)^{1/3} \tag{1}$$

where $m_p$ is the per-particle mass. For the ethylene premixed flame, sampling closer to the burner surface selected for smaller soot particles; sampling on the center line over the burner surface versus off-center may also affect the selected soot particle sizes. For the methane diffusion flame, increasing the fuel dilution (N$_2$) fraction generated smaller soot particles (Stipe et al., 2005). These variations in sampling and/or burner conditions likely led to some changes in the particle optical

and chemical properties (López-Yglesias et al., 2014). Thus, some of the size-dependent changes observed in particles smaller than $d_{p,VED} \sim 160$ nm, discussed below, are likely a result of real changes in the particle properties. The extreme case is in the comparison between the soot particles sampled at 5.1 cm (BC3+) and 20.3 cm (BC2) above the ethylene premixed flame burner surface.

**3.2   Particle Processing**

The soot particles were subjected to various physical and chemical processing and were selected according to their mobility diameter after processing but prior to sampling for optical and chemical property measurements. Particles size selected with no processing are termed nascent soot particles. For some experiments, nascent particles were passed through a thermal denuder in which

they were heated to 270°C for ~5 seconds. Such particles are termed nascent-denuded soot particles. (A summary of terminology is provided in Table S1.) For other experiments, nascent particles were first coated and then thermodenuded. Coating materials included either dioctyl sebacate (DOS, C$_x$H$_y$O$_x$; BC2, BC3, and BC3+), sulfuric acid (H$_2$SO$_4$; all studies), or secondary organic aerosol from α-pinene photooxidation (SOA; BC4). These particles are referred to as

coated-denuded. For DOS coatings, the nascent particles were first size selected, the monodisperse nascent particles were coated, and the coated particles were subsequently denuded. This is referred to as a forward-coating experiment. For either H$_2$SO$_4$ or SOA coatings, polydisperse nascent soot particles were first coated, then size selected, and finally denuded. This is referred to as a reverse-



coating experiment. This difference in methodology decreased the extent of homogenous nucleation of pure $H_2SO_4$ and SOA particles by providing additional soot surface area to act as a condensation sink, and any nucleated particles were further excluded during the size selection. However, the reverse method led to broader size and mass distributions in the denuded soot cores.

For BC2, BC3, and BC3+, soot particles were coated in a heated section of tubing containing either DOS or $H_2SO_4$, which cooled and condensed onto soot particles after exiting the heated section of tubing. For BC4 experiments, soot particles were coated with α-pinene SOA and $H_2SO_4$ that were generated in a Potential Aerosol Mass (PAM) oxidation flow reactor (Lambe et al., 2011). In the PAM reactor, $O_3$ was photolyzed by UV lamps at λ = 254 nm to produce $O(^1D)$ radicals, which

then reacted with water (RH=25-30%) to produce OH radicals. Low volatility products were formed via the reaction between OH and α-pinene, which condensed onto the soot particles. Similarly, $H_2SO_4$ was formed through the reaction of $SO_2$ and OH radicals.

Particles were size selected according to their mobility diameter ($d_m$) using a differential mobility analyser (DMA; TSI model 3080) with sheath-to-sample flow ratio of 5:1, yielding resolution of

~ ±20% of the set point in terms of mobility. During BC4, particles were mass-selected with a centrifugal particle mass analyser (CPMA; Cambustion Ltd.), in addition to being size-selected using a DMA. Particles sampled into the DMA or CPMA first passed through a neutralizer, which imparts an equilibrium charge distribution to the particles. The DMA selects particles according to their electrical mobility, which is dependent upon the number of charges per particle. Particles

with more than one charge are larger than those with a single charge, and the presence of these larger particles can confound interpretation of optical property measurements. Altering the burner and sampling conditions for both flame types minimized the number of these multiply-charged particles during the experiments, and in some cases their concentrations were effectively zero. However, for some experiments, their concentrations were non-zero. The method used to account

for the multiply-charged particles is discussed below.

### 3.3 Instrumentation

A wide range of instruments was employed to characterize the soot particle size, mass, composition and optical properties. Not all instruments were deployed for all studies, summarized in Table S2.



### 3.3.1 Size and Mass

Particle mobility size distributions were measured using a scanning mobility particle sizer (SMPS; TSI model 3936). Particles sampled into the SMPS were passed through a neutralizer, which imparts a new equilibrium charge distribution to the particles. This (re)neutralization step shifts

the charge distribution of the monodisperse particles imparted by the size-selection DMA to a new equilibrium state. For particles with mobility diameters 200-300 nm (typical of these studies), the number concentration ratio between particles with +1 charge to those with a greater number of charges selected by the DMA is ~2-2.5 at equilibrium. Thus, after (re)neutralization, the majority of the particles that were in a +2 state coming out of the DMA are sized in the SMPS in their +1

state. This allows for quantitative determination of the fraction of particles that had charge states of +1, +2, +3, etc., when selected by the DMA. The information is used to correct the optical measurements for contributions from larger, multiply-charged particles for BC2, BC3, and BC3+.

Particle mass distributions were measured using a CPMA. Particle number concentrations were measured using a condensation particle counter (CPC; TSI Inc.) and using a mixing CPC (MCPC,

BMI model l710). The two particle-counting instruments were placed at different points in the flow path to estimate and account for losses in the transfer lines between instruments. All CPCs number concentrations agreed within ± 5%.

### 3.3.2 Composition

Particle composition was characterized online using a soot particle aerosol mass spectrometer (SP-

AMS; Aerodyne Research, Inc.) (Onasch et al., 2012) and a compact time-of-flight aerosol mass spectrometer (c-ToF-AMS; Aerodyne Research, Inc.) (Canagaratna et al., 2007). The SP-AMS measured concentrations of refractory BC and the associated refractory and non-refractory coatings in BC-containing particles. The c-ToF-AMS measures concentrations of only non-refractory materials, which include particulate organic matter (POM) and some inorganic salts

($NH_4^+$, $SO_4^{2-}$, $NO_3^-$). Both instruments allow for determination of mass-weighted particle size distributions as characterized by the vacuum aerodynamic diameter ($d_{p,va}$). Particles were also collected on quartz-fiber filters for offline thermal-optical analysis (Chow et al., 2004), from which the relative abundances of OC and elemental carbon (EC) are determined.



During BC4, refractory black carbon mass was quantified with a single particle soot photometer (SP2; Droplet Measurement Technologies). The SP2 heats up the soot particles with an Nd:YAG laser and quantifies the incandescence emission as the particle evaporates. The incandescence signal from the instrument was calibrated using size-selected fullerene soot (Laborde et al., 2012).

The mass distributions from the SP2 were used to determine contributions from multiply-charged particles during BC4.

### 3.3.3  Optical properties

Particle optical properties were characterized using multiple instruments. Three different photoacoustic spectrometers (PAS) were used during the various studies to measure particulate

light absorption coefficients ($b_{abs}$). During BC2, a custom-built PAS from the National Oceanic and Atmospheric Administration (NOAA) operating at $\lambda = 532$ nm was deployed (Lack et al., 2006). Also, during BC2, a commercial 3-wavelength particle absorption soot spectrometer (PASS-3; DMT, Inc.) operating at $\lambda = 405$ nm, 532 nm and 781 nm was deployed by Los Alamos National Laboratory (LANL). During BC3, BC3+, and BC4, a custom-built PAS from the

University of California, Davis (UCD) operating at $\lambda = 405$ nm and 532 nm was deployed (Lack et al., 2011; Cappa et al., 2012). Light absorption was also measured during BC3, BC3+, and BC4 at $\lambda = 630$ nm using a commercial cavity attenuated phase-shift single scatter albedo spectrometer (CAPS PM$_{SSA}$; Aerodyne Research, Inc.). The CAPS PM$_{SSA}$ measures particulate light absorption as the difference between the measured light extinction ($b_{ext}$) and scattering ($b_{sca}$) coefficients,

whereas the PAS instruments measure light absorption directly. The CAPS extinction measurement has been previously described (Massoli et al., 2010). The CAPS PM$_{SSA}$ measures $b_{sca}$ using an integrating nephelometer, corrected for the finite viewing angle of the detector, i.e. truncation correction (Onasch et al., 2015). Light extinction coefficients were measured at $\lambda = 532$ nm during BC2 using the NOAA cavity ringdown (CRD) spectrometer, and at $\lambda = 405$ nm and 532

nm during BC3, BC3+, and BC4 using the UCD CRD spectrometer (Langridge et al., 2011). Light extinction coefficients were also determined during BC2 at $\lambda = 405$ nm, 532 nm and 781 nm using the PASS-3 instrument, which, like the CAPS PM$_{SSA}$, incorporates an integrating nephelometer to measure $b_{sca}$; $b_{ext}$ is determined as the sum of $b_{abs}$ and $b_{sca}$.

Absorption and extinction cross sections were determined for the size-selected particles as:



$$\sigma_{abs}(Mm^{-1}) = \frac{b_{abs}}{N_p} \tag{2}$$

$$\sigma_{ext}(Mm^{-1}) = \frac{b_{ext}}{N_p} \tag{3}$$

where $N_p$ is the measured particle number concentration. Additional parameters of interest that can be calculated from the measurements are the wavelength-dependent mass absorption and extinction coefficients, *MAC* and *MEC*, respectively, defined as:

$$MAC(m^2\ g^{-1}) = \frac{b_{abs}}{N_p \times m_p} \tag{4}$$

$$MEC(m^2\ g^{-1}) = \frac{b_{ext}}{N_p \times m_p} \tag{5}$$

where $m_p$ is the per particle mass. The combination of the absorption and extinction cross-sections allows for calculation of single scatter albedo, defined as:

$$SSA = 1 - \frac{\sigma_{abs}}{\sigma_{ext}}. \tag{6}$$

The *SSA* characterizes the fraction of extinction that is due to scattering. The wavelength-dependence of absorption is characterized by the semi-empirical parameter the absorption Ångström exponent, defined as:

$$AAE = \frac{\ln\left(\sigma_{abs,2}/\sigma_{abs,1}\right)}{\ln\left(\lambda_1/\lambda_2\right)} \tag{7}$$

where λ is wavelength and the subscript 1 and 2 indicate two different wavelengths.

## 4 Data Analysis

### 4.1 Mass Mobility Exponent

One measure of particle shape is the mass mobility exponent ($D_{f,m}$) (Cross et al., 2010):

$$m_p = C \cdot d_m{}^{D_{f\_m}} \tag{8}$$



where $C$ is the proportionality constant. Values of $D_{f,m}$ for atmospheric particles typically range from 2.0 to 3.0, with lower values being characteristic of more "lacy" soot and higher values of more compacted soot. $D_{f,m}$ values were obtained for nascent and coated-denuded soot particles by fitting a power law function to a graph of $m_p$ versus $d_m$.

## 4.2 Refractive Index Retrieval

Effective complex refractive index values (RI; $n = m + ki$) for the soot particles were determined using two methods: (1) spherical particle Mie Theory and (2) Rayleigh-Debye-Gans (RDG) approximation. For the refractive index, $m$ is the real component and $k$ is the imaginary component. In Mie Theory, particle cross-sections are calculated assuming spherical particles with a homogeneously uniform complex refractive index. The calculated cross-section is the product of the calculated absorption or extinction efficiency ($Q_{abs}$ or $Q_{ext}$) and the geometric particle cross-section ($= \pi \cdot [d_p/2]^2$, where $d_p$ is the particle diameter). Here, the particle diameter used in the calculations is the measured $d_{p,VED}$, as determined from the CPMA per-particle mass measurements and assuming a material density for BC of 1.8 g/cm$^3$. In the RDG approximation (Sorensen, 2001), the absorption cross-sections are calculated as:

$$\sigma_{abs,RDG} = N_{spherule}\sigma_{abs,spherule} \tag{9}$$

where $N_{spherule}$ is the number of spherules comprising a particle and $\sigma_{abs,spherule}$ is the absorption cross-section calculated for a single spherule using Mie theory. Here, we have assumed that $N_{spherule} = m_p/m_{spherule}$, where $m_p$ is the measured per-particle mass and $m_{spherule}$ is the mass of an individual spherule with $d_p = 20$ nm (ethylene premixed flame) (Cross et al., 2010) or $d_p = 37$ nm (methane diffusion flame) (Ghazi et al., 2013), again assuming a material density of 1.8 g/cm$^3$.

Optimal (best-fit), theory specific values of $m$ and $k$ were established by comparing the size-dependent observations of $\sigma_{abs}$ to calculations from either Mie theory or the RDG approximation by varying these parameters over the ranges $1.3 < m < 2.2$ and $0.1 < k < 1.5$ and determining the minimum value of the reduced chi-square statistic:



$$\chi^2_{red} = \frac{1}{N-1} \sum_j \left[ \frac{\left( \sigma^j_{abs} - \sigma^j_{abs,calc} \right)^2}{\varepsilon_{abs}^2} \right] \qquad (10)$$

where $\sigma^j_{abs}$ is the measured absorption cross-section, $\sigma^j_{abs,\,Mie}$ is the calculated absorption cross-section, $N$ is the number of points and $\varepsilon_{abs}$ is the estimated uncertainty, which accounts for both instrumental uncertainties plus the uncertainty associated with contributions from multiply-

charged particles. Individual fits were performed for each wavelength for each flame type. The methane diffusion flame soot data collected during different studies was treated as one dataset, whereas the ethylene premixed flame soot data collected during the different studies were kept separate due to the sampling differences described above.

## 4.3 Accounting for multiply charged particles

All absorption and extinction coefficient measurements were corrected for contributions from doubly-charged (Q2) particles. This correction leads to a decrease in the measured absorption and extinction coefficients. Here, an approach is taken where the fraction of absorption or extinction from Q2s ($f_{abs,Q2}$ or $f_{ext,Q2}$, respectively) is estimated from the measurements of the number fraction of singly and doubly charged particles ($f_{Q1}$ and $f_{Q2}$, respectively) determined from either the SMPS

(BC2, BC3, and BC3+) or the SP2 (BC4), the per-particle mass of singly and doubly charged particles ($m_{p,Q1}$ and $m_{p,Q2}$, respectively) and with an assumption of spherical particles (i.e. that $m_p = \rho(\pi/6)d_{p,VED}^3$), and that the $MAC$ and $MEC$ are size independent. The resulting expression for $f_{abs,Q2}$ is then:

$$f_{abs,Q2} = \frac{f_{Q2} \cdot m_{p,Q2} \cdot MAC}{f_{Q1} \cdot m_{p,Q1} \cdot MAC + f_{Q2} \cdot m_{p,Q2} \cdot MAC} = \frac{f_{Q2} \cdot m_{p,Q2}}{f_{Q1} \cdot m_{p,Q1} + (1-f_{Q1}) \cdot m_{p,Q2}} \sim \frac{f_{Q2} \cdot d_{p,VED,Q2}^3}{f_{Q1} \cdot d_{p,VED,Q1}^3 + (1-f_{Q1}) \cdot d_{p,VED,Q2}^3}$$

$$\qquad (11)$$

It is important to note that the final expression is independent of the absolute value of the $MAC$. A similar procedure was used to correct $MEC$ values. Although the $MAC$ (and $MEC$) may not be fundamentally size-independent, as assumed, the correction is relatively insensitive to differences in the $MAC$ between sizes. For the size range of interest, there is a relatively constant relationship between the diameters of the singly- and doubly-charged particles, with $d_{p,VED,Q2} \sim 1.6 d_{p,VED,Q1}$

corresponding to $m_{p,Q2} \sim 4 m_{p,Q1}$. Thus,





$$f_{abs,Q2} \sim \frac{4(1-f_{Q1})}{f_{Q1}+4(1-f_{Q1})} \,, \tag{12}$$

and the absorption by Q1 particles only is:

$$b_{abs,Q1} = b_{abs,obs}(1 - f_{abs,Q2}). \tag{13}$$

As an example, if $f_{Q1} = 0.95$ then $f_{abs,Q2} = 0.17$, and the sensitivity of $f_{abs,Q2}$ to uncertainties in $f_{Q1}$

is $\delta f_{abs,Q2} \sim 3 \cdot \delta f_{Q1}$, as determined by sensitivity calculations. We estimate the uncertainty in $f_{Q1}$ is

0.01, which contributes ~3% to the uncertainty in $b_{abs,Q1}$. (If the $MAC$ were not constant, but were

instead 20% larger for Q2s compared to Q1s, then the correction would increase to 0.20 from

0.17.) For most runs, the Q2 contribution was small, being 5% or less.

## 5 Results and Discussion

Results from the methane diffusion flame and from the two different ethylene premixed flame

experiments are presented and discussed separately. For each flame type or configuration, the

coated-denuded data for all coating types are considered together.

### 5.1 Soot optical properties from the methane diffusion flame

#### 5.1.1 *MAC values of nascent and coated-denuded soot*

Average and median $MAC$ values were determined for BC particles from the methane diffusion

flame with $x > 0.9$ (corresponding to a $d_{p,VED} = 160$ nm at $\lambda = 532$ nm; Table S3). There are no

systematic differences in the $MAC$ values between nascent and nascent-denuded ($D_{f,m} = 2.16 \pm$

0.1) and coated-denuded ($D_{f,m} = 2.64 \pm 0.1$) soot at any wavelength for this range of $x$ despite some

degree of collapse for the thickly coated-denuded particles (Figure 1 and Figure S2). The collapse

was presumably due to the effect of evaporation or condensation of the coating material, and not

due to the denuding process alone (Bhandari et al., 2016). The observed $D_{f,m}$-independence of the

$MAC(x>0.9)$ is consistent with Radney et al. (2014). This contrasts, however, with modelling

studies that use non-Mie based methods that can account for particle shape effects, which indicate

"lacy" soot ($D_{f,m} = 1.8$) is more absorbing than "compact" soot ($D_{f,m} = 2.4$) (Kahnert and

Devasthale, 2011; Scarnato et al., 2013). In the calculations, the compact soot particles are less



absorbing because the inner-most spherules are "shielded" by the outer-most spherules. It is possible that the extent of collapse here was insufficient to lead to substantial "shielding" in our experiments. Regardless, given the similarity of the observed nascent and coated-denuded particle cross-sections, they have been recombined into a single dataset in what follows.

### 5.1.2   RI values calculated from Mie Theory and the RDG approximation

The $\sigma_{abs}$ have been fit separately using Mie theory and the RDG approximation to determine optimal, theory-specific effective complex *RI* values. The observations and best fits are shown in Figure 2 at the three wavelengths ($\lambda$ =405 nm, 532 nm, and 630 nm), and the derived optimal wavelength-, flame- and theory-specific values are reported in Table 1. The quality of the best fit obtained is dependent upon both theory and wavelength considered. First, fits performed using Mie theory tend to give reasonably well-defined minima in the calculated $\chi^2_{red}$, indicating that the optimal *m* and *k* are unique (Figure S3). In contrast, fits performed using the RDG approximation do not give a unique set of *m* and *k*, but instead a band of [*m,k*] pairs that describe the data equally well (Figure S4). Since RDG fits are non-unique, optimal *k* values are reported at all wavelengths for a fixed value of *m* = 1.80.

There are additional differences between Mie and RDG beyond the uniqueness of the derived optimal *RI*. At $\lambda$ = 405 nm, Mie theory provides a poor fit to the $\sigma_{abs}$ when the fit is performed using data over the entire size range sampled. In particular, at $\lambda$ = 405 nm the $\sigma_{abs}$ from Mie theory are overestimated below $d_{p,VED}$ ~120 nm ($x$ ~ 0.9) and underestimated for larger sizes (Figure 2A). At $\lambda$ = 532 nm, this deviation also occurs at $x$ ~ 0.9 ($d_{p,VED}$ > 160 nm) corresponding to larger size particles. Compared to 405 nm, the overestimate at small $x$ and underestimate at larger $x$ for 532 nm is smaller. At $\lambda$ = 630 nm, the Mie theory fit compares well with the observations at $x$ < 0.9 ($d_{p,VED}$ ~ 180 nm) and with some deviation observed at larger sizes. When the fits are restricted to $x$ < 0.9, a reasonable fit using Mie theory is obtained at all wavelengths over this size range. However, these constrained Mie fits extrapolated to larger sizes ($x$ > 0.9) still underestimate the observed absorption. For RDG, generally good fits are obtained at all wavelengths and across all sizes, although the RDG-calculated $\sigma_{abs}$ tend to overestimate the observation at smaller sizes below $d_{p,VED}$ ~100 nm.





It is important to note that the *RI* values listed in Table 1 are theory and property specific. This means that the Mie-derived *RI* values are not appropriate for use with the RDG approximation, and vice versa. Additionally, they must be used assuming a material density of 1.8 g/cm$^3$, since this value was used to convert $m_p$ to $d_{p,VED}$ or $N_{spherule}$. If, for example, a smaller density were used

with these RI values then the particles would have substantially higher *MAC*s. Of note is that both the real and imaginary *RI*s from Mie theory are larger than *RI* values that are commonly used in global climate models, (Bond et al., 2013), including the *RI* that is often considered the currently recommended value (1.95 – 0.79*i*) (Bond and Bergstrom, 2006).

### 5.1.3    *Comparison of measured and calculated MAC values*

Another way to look at the extent to which Mie theory or the RDG approximation can reproduce the observations is to compare the observed and calculated *MAC* values as a function of particle size (or *x*), rather than the $\sigma_{abs}$ versus size relationship (as in Figure 2). Although the *MAC* is related directly to the $\sigma_{abs}$, it is nonetheless useful to consider the *MAC* values because they vary over a much narrower range than do the $\sigma_{abs}$. The dependence of the *MAC* on $d_{p,VED}$ and size parameter

for both the observations and the models are shown in Figure 3. The observed *MAC* values generally increase with $d_{p,VED}$ or size parameter at all wavelengths up to around $d_{p,VED} \sim 160$ nm, above which they plateaus and are approximately constant. The ranges (minimum and maximum) of binned observed *MAC*s are provided in Table S4.

    The *MAC* values observed here (Table S3) are substantially larger than the value of $5.7 \pm 0.8$ m$^2$

g$^{-1}$ at $\lambda = 405$ nm reported by Radney et al. (2014), who use a Santoro-type burner (i.e., co-flow diffusion flame). They did not report any notable size dependence to their *MAC* values. Our *MAC* values (especially for $x > 0.5$) compare well with the value of 8.16 m$^2$g$^{-1}$ at $\lambda = 532$ reported by You et al. (2016) (extrapolated from $7.89 \pm 0.25$ m$^2$g$^{-1}$ at $\lambda = 550$ nm) for soot particles generated from the combustion of organic fuel stock over the $d_{p,VED}$ range $\sim 80$ nm to $\sim 210$ nm. They also

compare favorably to the range of values 7.2 to 8.5 m$^2$g$^{-1}$ reported for $\lambda = 532$ nm for $d_{p,VED} \sim 100$ nm observed in Saliba et al. (2016) for particles generated from a cookstove. Some particle size dependence was reported by Khalizov et al. (2009), who used propane with a Santoro-type burner, with reported *MAC*s at $\lambda = 532$ nm of $6.7 \pm 0.7$ m$^2$g$^{-1}$ for $d_m = 155$ nm particles and $8.7 \pm 0.1$ m$^2$g$^{-1}$ for $d_m = 320$ nm particles. This general behavior was also observed for soot particles generated





from a methane diffusion flame in Dastanpour et al. (2017), with $MAC$s values reported at $\lambda = 660$ nm of ~5 $m^2$ $g^{-1}$ for $d_{p,VED} = 50$ nm and ~7 $m^2$ $g^{-1}$ for $d_{p,VED} = 100$ nm.

The observations are compared with the calculated $MAC$ values, based on the fits from Figure 2. $MAC$ values from the RDG approximation are independent of $x$, as the particle $MAC$ is equal to the $MAC$ of the individual spherules making up the particle. Here, the observed $MAC$ values in the plateau (large $x$) regime correspond reasonably well with the $MAC$s as calculated from RDG values when the optimal RI values are used. The constant RDG $MAC$ value at $\lambda = 532$ nm (= 8.8 $m^2$ $g^{-1}$) is slightly larger than the often suggested value for atmospheric BC by Bond and Bergstrom (2006) of 7.75 ± 1.2 $m^2$ $g^{-1}$ (extrapolated from 7.5 at $\lambda = 550$ nm using 1 for the $AAE$) and is identical to that reported for soot from a Santoro-type diffusion burner operating on propane (Zhang et al., 2008). However, the $MAC$s predicted from RDG overestimate the observed $MAC$s at $x < 0.90$, since the RDG fits are weighted by the greater number of data points at $x > 0.9$ where $MAC$s are approximately constant.

For Mie theory, calculated $MAC$ values for highly absorbing substances, such as BC, have a characteristic shape where the $MAC$ is constant up to $x \sim 0.2$, increases monotonically by ~40% until $x \sim 0.9$, and then decreases rapidly towards larger $x$. The Mie theory curves calculated here reproduce the observed values at $x < 0.90$ (especially for $\lambda$=532 nm and 630 nm), but substantially underestimated $MAC$s at $x > 0.90$. This facilitates understanding of the Mie model underestimate of $\sigma_{abs}$ at large $x$ (Figure 2). Above $x \sim 0.9$ the calculated Mie $MAC$ declines with $x$ for all wavelengths, but the observations indicate that the $MAC$ is constant. Because $x$ occurs at smaller $d_{p,VED}$ for shorter wavelengths than for longer wavelengths, the model-measurement difference in both $\sigma_{abs}$ and $MAC$ is more noticeable at 405 nm than it is at 532 nm, which is more noticeable than at 630 nm. This is a consequence of a greater number of the data points at $x > 0.9$ at 405 nm, past the peak in the Mie-calculated $MAC$.

The reasonable correspondence between the observed and Mie-calculated $MAC$ at smaller sizes is, however, somewhat surprising given that Mie assumes spherical particles, yet the particles are not spherical. One potential reason for the observed dependence of the $MAC$ on particle size is that the chemical and optical properties of the particles change with size, and the different chemical composition coincidentally improves agreement with Mie theory. For the diffusion flame, changes





in the particle size distribution were induced by changing the amount of dilution nitrogen in the sheath flow. This can influence the maturity of the soot, where maturity refers, in general, to the hydrogen content of the soot and is known to influence soot absorption (López-Yglesias et al., 2014). The observed increase in *MAC* with $d_{p,VED}$ here exhibits some wavelength dependence,

which could reflect differences in the sensitivity of the *MAC* to maturation. The observed differences between the *MAC*s observed at the smallest $x$ and the maximum *MAC* values were 21%, 41% and 37% for $\lambda = 405$ nm, $\lambda = 532$ nm and $\lambda = 630$ nm, respectively. However, one additional difference between the wavelengths is that at $\lambda = 405$ nm there is a small increase in *MAC* going from $d_{p,VED} \sim 60$ nm ($x_{405nm} = 0.6$) to $\sim 100$ nm ($x = 0.9$) after which the *MAC* is constant,

while at $\lambda = 630$ nm there is much more of a continuous increase in the *MAC* up to larger particle sizes. This could indicate that, at some point, further changes in the soot maturity (composition) have no influence at short wavelengths, but do at longer wavelengths. As a complementary explanation, Dastanpour et al. (2017) observed an increase in primary spherule size with overall particle size for methane diffusion flame generated soot. They attribute the increase in *MAC* with

$d_{p,VED}$ to changes in the internal structure and/or the degree of graphitization that occur with changes in spherule size.

The observation of constant *MAC* values for mature soot ($x > 0.9$ or $d_{p,VED} \sim 160$ nm at $\lambda = 532$ nm) is an important result in the context of how BC is commonly treated in climate models. Most climate models simulate the optical properties of BC using spherical particle Mie theory. The

observations indicate that Mie theory will likely underestimate the absorption by BC for particles with $x > 0.9$ because, when the particles are sufficiently absorbing, attenuation of light by the outer layers of the (spherical) causes the mass in the center of the particle to not interact with the electromagnetic field (Bond and Bergstrom, 2006; Kahnert and Devasthale, 2011). This suggests that the RDG approximation, or even an assumption of a constant *MAC*, may provide a more

accurate representation of BC absorption than Mie theory in climate models, at least for uncoated BC. This conclusion is independent of soot maturity, as the fall off in the *MAC* with increasing size for Mie theory occurs for all strongly absorbing particles. Although atmospheric BC particles are predominately generated through combustion of fossil fuels or through biomass burning (Bond et al., 2013), flame-generated BC particles have been shown to be a suitable proxy for atmospheric

BC particles, both in terms of chemical bonding and structural properties (Hopkins et al., 2007;





Slowik et al., 2007). The absolute values of the derived *RI* may be different for diesel or biomass BC particles, but it can be reasonably assumed that Mie theory does not reproduce the behavior of atmospheric BC particles.

The discrepancy between Mie theory and the observations is both size and wavelength dependent.
Consequently, the extent to which the true absorption by BC is underestimated by a given atmospheric model due to inappropriate use of Mie theory will depend importantly on the assumed size distribution, both the position and the width, and the wavelength. Using the effective *RI* values determined here, we estimate that absorption is underestimated by around 20-40% when Mie theory is used with reasonable BC size distributions. The underestimate in absorption from the use
of Mie theory will be even larger if non-theory-specific (typically lower) imaginary *RI* values are used, such as that suggested by Hess et al. (1998) or Bond and Bergstrom (2006), as discussed by Stier et al. (2007). Consider that the maximum *MAC* predicted using the Hess et al. (1998) *RI* (=1.75-0.44$i$) from Mie theory is only 3.8 m$^2$ g$^{-1}$

### 5.1.4  *MEC and SSA values of nascent and coated-denuded soot*

Measurement of $b_{ext}$ and the *MEC* were made in addition to the $b_{abs}$ and *MAC* measurements (Figure 3). Above, the *RI* fitting was performed using only the absorption measurements, in part because calculation of extinction using RDG requires additional assumptions regarding the particle shape. Nonetheless, it is informative to compare the $\sigma_{ext}$ and *MEC* observations to Mie theory calculations since the overall climate impacts of BC depend on both absorption and scattering. As
with the *MAC* values, the observed *MEC* values also increase with $x$ up to 0.90 (or $d_{p,VED}$ up to 160 nm), after which point they are relatively constant (Figure 3). The Mie theory *MEC*s calculated using the *RI*s determined by fitting the absorption measurements (Table 1) agree reasonably well with observations when $x < 0.9$ (using the fits that were constrained to this range), but, as with absorption, Mie theory underestimates the *MEC* above $x = 0.9$. For a given particle size, there is
somewhat greater scatter in the observed *MEC*s than in the *MAC*s. This is likely a result of the scattering being more sensitive to the shape of the soot particles than is absorption and that the nascent and coated-denuded particle results are combined here.



Given this, the dependence of the *SSA* on particle size is considered separately for the nascent (more lacy) and coated-denuded (more compact) particles. The coated-denuded particle *SSA*s increase with $d_{p,\mathrm{VED}}$, most noticeably for $d_{p,\mathrm{VED}} > 100$ nm, up until $d_{p,\mathrm{VED}} \sim 160$ nm. Above this size the SSA values are approximately constant at a value of $\sim 0.30$ (Figure 4A). (Results at $\lambda = 532$

nm are shown in Figure 4A, but there is a strong correlation between *SSA* at 532 nm and at 405 nm or 630 nm; Figure S5). In contrast, the nascent SSAs increase slightly from $d_{p,\mathrm{VED}} \sim 50$ nm to 80 nm, but above 80 nm are approximately constant at $\sim 0.20$. This demonstrates that particle collapse leads to an increase in the SSA for BC, consistent with Radney et al. (2014). This behavior is also consistent with modelling studies, which have predicted that compact agglomerates exhibit

higher SSA values than lacy agglomerates, with an absolute increase of $\Delta SSA \sim 0.1$ (Scarnato et al., 2013) or by a factor of $1.2 - 2.2$, depending on the extent of compaction (China et al., 2015a; China et al., 2015b). The increase upon collapse is attributed to the stronger scattering and electromagnetic coupling between spherules in compact aggregates. Here, the difference between the SSA for nascent and coated-denuded soot increases somewhat with particle size, which may

result from changes in soot maturity with size. The *SSA*s from this study compare reasonably well to values reported previously at visible wavelengths. For example, Saliba et al. (2016) report *SSA* = 0.16 to 0.26 for nascent soot emitted from a cookstove; Schnaiter et al. (2003);(2006) report *SSA* = 0.2 to 0.3 for soot from a propane diffusion flame, SSA = 0.18 to 0.25 for kerosene-derived soot, and SSA = 0.1 to 0.25 for methane diffusion flame soot; and Sharma et al. (2013) report *SSA*

= 0.18 to 0.25 for soot generated from a kerosene lamp. However, the values reported here, are much smaller than the value of 0.5 reported in Radney et al. (2014). The Mie-theory calculated *SSA* values are similar to observations, although show a somewhat stronger increase with size and seem to plateau at larger *SSA* values at large sizes compared to observations.

### 5.1.5 Absorption Ångström Exponent (AAE) of nascent and coated-denuded soot

The wavelength dependence of absorption has been considered by calculating the *AAE* using the measurements at $\lambda = 405$ nm and 532 nm (Figure 4B). The nascent *AAE*s at larger particle sizes are slightly larger than the coated-denuded *AAE*s, suggesting that particle collapse leads to a slight decrease in the wavelength dependence of absorption. The average *AAE* was $1.38 \pm 0.36$ (N = 85) and $1.10 \pm 0.37$ (N = 135) for nascent and coated-denuded particles, respectively. There is some





indication that *AAE* decreases with particle size. This may again be the result of the soot maturity increasing (and the composition changing) with size. The *AAE* values from Mie theory, based on the best-fit *RI* values determined above, exhibit an increase to $x \sim 0.5$ where they peak and then decrease sharply. This predicted decrease is inconsistent with the observations. The *AAE*s

calculated using the RDG approximation from the best-fit RI values are constant.

BC is commonly assumed to have an *AAE* = 1 (Bergstrom, 1973). The measurements here are consistent with this expectation for the collapsed (coated-denuded) particles, but the nascent particles give an *AAE* that is somewhat larger than 1. The observed *AAE* values are similar to results from previous studies examining either freshly emitted soot particles or soot particles

containing very little organic material (Bergstrom et al., 2002; Schnaiter et al., 2003; Kirchstetter et al., 2004; Schnaiter et al., 2006; Sharma et al., 2013; You et al., 2016).

### 5.2    Optical properties of BC from the ethylene flat-burner flames

#### 5.2.1    *Results from BC2: Sampling high above the burner surface*

During BC2, the ethylene flame was sampled at a height of $\sim$20.3 cm above the surface. At this

height, the particles were likely reasonably mature, at least relative to sampling that was performed further into the flame, as was done in BC3+. Particle optical properties were quantified at $\lambda = 405$ nm, 532 nm, and 781 nm, with two independent measurements at 532 nm considered (NOAA and PASS-3). As with the methane diffusion flame, the data were fit using Mie theory and the RDG approximation to determine optimal, theory-specific, wavelength-dependent RI values (Table 2).

The soot particles from this flame had, overall, a greater amount of intrinsic organic carbon associated with them compared to the particles from the methane diffusion flame. As such, denuding even of the nascent particles led to changes in the optical properties and particle masses. Thus, the nascent and denuded particles are considered separately, and we focus on the denuded particles. The range of particle sizes considered was also overall smaller than that for the methane

diffusion flame.

The retrieval of effective refractive indices for this flame using Mie theory resulted in a good fit to $\sigma_{abs}$ versus $d_{p,\text{VED}}$ for all $\lambda$ and $d_{p,\text{VED}}$ (Figure 5A, D,G, and J, and Figures S6-S7). This difference from the methane diffusion flame is in large part due to the more restricted size range encountered





here, with BC particles only up to $d_{p,\text{VED}}$ = 160 nm used. The RDG approximation yielded a reasonably good fit across all particle sizes for the ethylene particles, although with some overestimation at smaller sizes. The *MAC* values tended to increase with particle size or size parameter, most obviously at 532 nm where the most data points are available (Figure 5B, E, H,

and K). The range of binned *MAC*s shown in Figure 5 are listed in Table S4. The *MAC* values determined using the two PAS instruments at 532 nm differ somewhat, with the NOAA PAS *MAC*s slightly larger than PASS-3 *MAC*s, although the differences are within the measurement uncertainties. Most likely, this instrument difference stems from differences in calibration methods.

The observed *MAC* values tend to be larger for the denuded particles than for the nascent particles, most likely due to less-absorbing organics present in nascent soot that contribute ~25% of the particle mass (Cross et al., 2010). The *MAC* values at λ = 405 nm and 532 nm for the denuded BC2 ethylene flame soot are comparable, within uncertainty, to the *MAC* values for the methane diffusion flame soot for the particles with $d_{p,\text{VED}}$ ~ 150 nm. This indicates that the BC from these

two flames is similarly absorbing in nature.

The impact of particle morphology on the *SSA* and *AAE* is considered by comparing the results for nascent-denuded particles with coated-denuded particles (Figure 6). Nascent particles are excluded because the presence of intrinsic organics can increase *AAE* if the organic material contains brown carbon and can increase the *SSA* independent of the underlying BC morphology. The $D_{\text{f,m}}$ values

were $2.12 \pm 0.06$, $2.49 \pm 0.07$, and $2.17 \pm 0.06$ for nascent-denuded, sulfuric acid coated-denuded, and DOS coated-denuded soot, respectively (Cross et al., 2010). The nascent-denuded and DOS coated-denuded particles have *SSA* values close to 0, whereas the sulfuric acid coated-denuded particles have *SSA* values closer to 0.15 at *x* > 0.5, consistent with particle collapse leading to an increase in *SSA* (Figure 6A). The *SSA* values for the nascent-denuded particles from this flame are

smaller than the *SSA* values for the nascent particles from the methane diffusion flame. The reason for this is not clear, but is likely related to differences in the particle shapes and/or the sizes of the spherules; the $D_{\text{f,m}}$ for the nascent-denuded particles here ($D_{\text{f,m}}$ =2.12) is slightly smaller than for the particles from the methane diffusion flame ($D_{\text{f,m}}$ = 2.28) (Figure S8). The sulfuric acid coated-denuded particle *SSA* values from the BC2 ethylene flame are also lower than the methane



diffusion flame coated-denuded *SSA* values for a given size (Figure 6A), despite having similar $D_{f,m}$ values.

The *AAE* values (using the $\lambda = 405$ nm and 532 nm pair) for nascent-denuded and coated-denuded particles are generally similar when compared over the same size range (Figure 6B). However,

the sulfuric acid coated-denuded particle *AAE* values may be slightly smaller than for the nascent-denuded or DOS coated-denuded particles; all are close to unity at $x \sim 0.8$. This indicates that changes in morphology do not lead to substantial changes in *AAE*. The *AAE*s for the coated-denuded particles decrease strongly with $d_{p,VED}$, with the mean *AAE* ~1.6 for the smallest particles ($x = 0.5$) and *AAE* mean ~ 0.5 for the largest particles ($x = 0.9$). Values below 1 contrast with the

methane data, but have been observed in a few previous studies (Clarke et al., 2007; Hadley et al., 2008; Lack et al., 2008). While such a decrease with size is consistent with Mie theory predictions, given the *MAC* results it seems more likely that the size dependence of the *AAE* is related to changes in the soot maturity with particle size. To the extent that the larger particles, which tend to have larger *MAC* values, are reflective of more mature soot this suggests that mature soot from

this flame type has an *AAE*<1.

### 5.2.2   Results from BC3+: Sampling near the burner surface

During BC3+, the particles were sampled at variable heights above the burner surface to select for particles in different size ranges, but most often they were sampled from relatively close to the burner surface (5.1 cm) compared to BC2 sampling conditions (20.3 cm). It is likely that these

different sampling conditions gave rise to particles with different chemical properties. Multiple studies have shown changes in soot maturity and soot optical properties as a function of sampling height in ethylene premixed flames, though at a distance significantly closer to the flame front than sampled here (Migliorini et al., 2011; Olofsson et al., 2015). The particles during BC3+ had very little, if any, intrinsic organic carbon, unlike the BC2 particles that were ~25% organic by mass.

However, the small organic content for BC3+ was likely a consequence of the use of a hot sampling line prior to dilution. Consequently, the optical properties of both nascent and denuded ethylene soot particles from BC3+ (sampled close to the burner) differ substantially from the BC2 particles (sampled well above the burner). Average *MAC* values over various size ranges are listed in Table S5. In general, the *MAC*s for BC3+ ethylene particles are similar to BC2 particles at $\lambda = 405$ nm,



whereas the λ = 532 nm *MAC* for BC3+ particles are smaller than those for BC2 particles. (Measurements at λ = 630 nm were not made during BC2 nor were measurements at 781 nm made during BC3+.) At $d_{p,VED}$ > 70 nm, the λ = 405 nm *MAC* values were approximately constant with increasing $d_{p,VED}$ (Figure S9). The behaviour is consistent with methane diffusion flame

observations, but the constant *MAC* seems to occur at a lower $d_{p,VED}$. The number of available data point available for the BC3+ ethylene particles is limited, making conclusions regarding size-dependence of properties somewhat tenuous. The *AAE* for BC3+ ethylene particles are reasonably independent of particle size (Figure S10). The average value of $AAE_{405nm-532nm}$ = 2.01 ± 0.21 for $x$ > 0.5, which is higher than observed for the methane diffusion flame for this range of $x$ (=1.18 ±

0.35). These observations indicate that differences in sampling and soot maturity result in different optical properties. Previous studies have observed differences in optical properties, chemical composition, and primary spherule size for different flame sampling heights (Bladh et al., 2011; Migliorini et al., 2011; Olofsson et al., 2015). Absorption by less mature soot appears to decrease more rapidly with wavelength than for more mature soot, such that the *MAC* values in the mid-

visible (e.g. λ = 532 nm and 630 nm) are lower for less mature soot. These wavelength-dependent optical results appear to match trends observed previously using active remote sensing techniques to characterize particles within flames (Migliorini et al., 2011; Olofsson et al., 2015). The extent to which this conclusion can be generalized will require further investigation.

## 6    Conclusion

Light absorption and extinction cross-sections were measured for nascent, denuded, and coated-denuded soot particles that were produced from two different flame types that operated on different fuels (methane or ethylene). These measurements were used in conjunction with particle mass and size measurements to determine various intensive optical properties (e.g. *MAC*, *SSA* and *AAE*) for uncoated BC particles in the size range 50 nm < $d_{p,VED}$ < 210 nm (corresponding to 0.1 fg < $m_p$ <

5 fg). The optical properties varied somewhat with particle size, most likely due to changes in the chemical nature (i.e. maturity) of the BC that results from variations in the combustion and sampling conditions. However, for larger, mature particles, corresponding to those with $d_{p,VED}$ > ~160 nm, the observed intensive properties were generally size independent. The observed *MAC* values for BC, measured over multiple studies, are independent of particle collapse and thus



provide evidence that absorption by soot of a given maturity level is dictated primarily by the individual spherules and is thus largely size- and shape-independent. The observed *MAC* values are also larger than the recommended value of Bond and Bergstrom (2006), i.e. 8.6 m$^2$ g$^{-1}$ versus 7.75 m$^2$ g$^{-1}$ at 532 nm.

The observations serve as the basis for determination of wavelength- and theory-specific effective complex refractive indices, using both Mie theory and the RDG approximation. With Mie theory, good fits were only obtained for size parameters smaller than ~0.90 (corresponding to $d_{p,VED}$ ~160 nm for $\lambda$ = 532 nm). Above this size, Mie theory predicts a sharp decrease in the *MAC* while the observed *MAC* are constant. Thus, Mie theory systematically under-predicts the observed

absorption for $x > 0.9$ and a good fit is not possible. This is because with Mie theory when the particles are sufficiently absorbing and large, light is attenuated by the outer layers of the (spherical) particle such that the mass in the center of the particle does not interact efficiently with the electromagnetic field (Bond and Bergstrom, 2006; Kahnert and Devasthale, 2011).

Our analysis has important implications for the calculation of absorption in atmospheric models.

Atmospheric models that use Mie theory, which is the majority, likely underestimate the actual absorption by uncoated BC whether or not theory-specific *RI* values are used; the magnitude of the underestimation will depend on the assumed BC size distributions, increasing with increasing $d_{p,VED}$ and size distribution width. This may be especially important to consider for simulation of absorption by BC particles from biomass burning, which are larger than those from urban sources

(Schwarz et al., 2008). The underestimate of absorption by Mie theory will likely be even larger if non-theory specific RI values that inherently underestimate *MAC* values are used, which includes some of the more commonly used RI values.

Overall, our results demonstrate that either an assumption of a constant *MAC* or the use of the RDG approximation with theory-specific *RI* values (which are equivalent) in atmospheric models

are likely to provide for more accurate representation of absolute absorption by uncoated BC than does Mie theory. Further work will be necessary to understand how these results for uncoated BC will impact calculations of absorption by coated BC, for which absorption can be enhanced. However, our results suggest that the absolute absorption by coated particles will be similarly



underestimated if core-shell Mie theory is used, regardless of the accuracy of the absorption enhancement calculation.

## 7   Data Availability

The data associated with this paper are archived at the UC Davis DASH data repository and are

available for download from https://doi.org/10.25338/B8JP4V.

## 8   Acknowledgements

The BC2 study was supported by DOE grant No. DE-FG02-05ER63995 and the Atmospheric Chemistry Program of the National Science Foundation grants No. ATM-0525355.  The BC3 and BC3+ studies were supported by the DOE ASR program grant DE-SC0006980, the EPA STAR

Program grant R835033, and the NSF grant ATM-0854916. The BC4 study was supported by DOE grant DE-SC0011935 and the NSF grants AGS-1244918 (Boston College) and AGS-1244999 (Aerodyne).  LANL thanks U.S. Department of Energy's Atmospheric System Research, an Office of Science, Office of Biological and Environmental Research program for funding

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





## 10 Tables and Figures

**Table 1.** Refractive indices for methane diffusion flame soot retrieved via fitting Mie Theory and the RDG approximation to the $\sigma_{abs}$ observations. Nascent and denuded data are combined (see text for details).

| Study | Method | $\lambda$ (nm) | Instrument | $n = m + ki$ | $MAC_{peak}$ (m²/g) | $dp$ VED, peak (nm) /Size Param.[@] | Number of data points |
|---|---|---|---|---|---|---|---|
| BC3(+), BC4 | Mie[%] | 405 | UCD CRD-PAS | 1.29 + 1.29i[#] | 18.72 (±6.57)[#] | 61/0.47 | 225 |
| BC3(+), BC4 | Mie$_{x<0.9}$ | 405 | UCD CRD-PAS | 2.31 + 1.26i | 12.21 (±2.18) | 108/0.84 | 40 |
| BC3(+), BC4 | RDG[$,*] | 405 | UCD CRD-PAS | 1.80 + 1.39i | 11.91 (±1.97) | | 225 |
| BC3(+), BC4 | Mie | 532 | UCD CRD-PAS | 2.24 + 1.19i[#] | 9.13 (±3.03)[#] | 142/0.84 | 219 |
| BC3(+), BC4 | Mie$_{x<0.9}$ | 532 | UCD CRD-PAS | 1.96 + 1.01i | 8.68 (±1.08) | 137/0.82 | 82 |
| BC3(+), BC4 | RDG | 532 | UCD CRD-PAS | 1.80 + 1.43i | 8.81 (±1.46) | | 219 |
| BC3(+), BC4 | Mie | 630 | Aerodyne CAPS | 2.14 + 0.94i[#] | 6.93 (±0.20)[#] | 181/0.90 | 169 |
| BC3(+), BC4 | Mie$_{x<0.9}$ | 630 | Aerodyne CAPS | 2.01 + 0.89i | 6.73 (±0.24) | 177/0.88 | 134 |
| BC3(+), BC4 | RDG | 630 | Aerodyne CAPS | 1.80 + 1.13i | 6.51 (±1.26) | | 169 |

[%] Uncertainties in the $MAC$ are 1σ from the reduced $\chi^2$ fit. See supplementary figures S3 and S4.

[$] The $n$ values from the RDG method are non-unique. Therefore, uncertainty estimates from this work are not available. See text for details.

[@] VED and the size parameter ($x = \pi d_p/\lambda$) where the peak $MAC$ occurs

[*] There are many degenerate RI combinations that give similar quality fit to RDG theory. Thus, a value of 1.80 was chosen for the effective real refractive index.

[#] These values are given for reference purposes only, but should not be used due to inability to fit data well (see text for details).





**Table 2.** Theory-specific effective refractive indices for ethylene premixed flame soot from BC2 and BC3+ retrieved via fitting Mie Theory and the RDG approximation to the $\sigma_{abs}$ observations. Nascent and denuded experiments are considered separately.

| Study[#] | Soot Type | Method | $\lambda$ (nm) | Instrument | $n = m + ki$ | $MAC_{peak}$ (m$^2$/g) | $dp_{VED, peak}$ (nm) /Size Param.[@] | No. of data points |
|---|---|---|---|---|---|---|---|---|
| BC2 | Nascent | Mie[%] | 405 | PASS-3 | 2.21 + 0.86i | 10.45 (±2.06) | 123/0.95 | 36 |
| BC2 | Denuded | Mie | 405 | PASS-3 | 2.19 + 0.91i | 10.68 (±1.97) | 119/0.92 | 30 |
| BC2 | Nascent | RDG[$,^] | 405 | PASS-3 | 1.80 + 1.13i | 10.06 (±2.22) | | 36 |
| BC2 | Denuded | RDG | 405 | PASS-3 | 1.80 + 1.18i | 10.32 (±2.56) | | 30 |
| BC2 | Nascent | Mie | 532 | PASS-3 | 2.13 + 0.64i | 6.92 (±0.78) | 178/1.07 | 36 |
| BC2 | Nascent | Mie | 532 | NOAA CRD-PAS | 2.39 + 0.79i | 8.02 (±0.13) | 169/0.99 | 43 |
| BC2 | Denuded | Mie | 532 | PASS3 | 1.96 + 0.83i | 7.70 (±0.90) | 150/0.89 | 31 |
| BC2 | Denuded | Mie | 532 | NOAA CRD-PAS | 2.56 + 1.11i | 9.00 (±0.80) | 152/0.90 | 46 |
| BC2 | Nascent | RDG | 532 | PASS-3 | 1.80 + 0.78i | 6.92 (±0.43) | | 36 |
| BC2 | Nascent | RDG | 532 | NOAA CRD-PAS | 1.80 + 0.85i | 6.16 (±1.75) | | 43 |
| BC2 | Denuded | RDG | 532 | PASS-3 | 1.80 + 1.08i | 7.35 (±1.69) | | 31 |
| BC2 | Denuded | RDG | 532 | CRD-PAS | 1.80 + 1.13i | 7.55 (±0.68) | | 46 |
| BC2 | Nascent | Mie | 781 | PASS-3 | 2.16 + 0.76i | 5.10 (±0.40) | 250/0.96 | 31 |
| BC2 | Denuded | Mie | 781 | PASS-3 | 2.84 + 0.74i | 6.20 (±0.20) | 239/0.95 | 36 |
| BC2 | Nascent | RDG | 781 | PASS-3 | 1.80 + 0.50i | 2.59 (±0.33) | | 36 |
| BC2 | Denuded | RDG | 781 | PASS-3 | 1.80 + 0.73i | 3.64 (±0.81) | | 31 |
| BC3+ | Denuded | Mie | 405 | UCD CRD-PAS | 2.11 + 1.03i | 11.09 (±2.82) | 110/0.85 | 27 |
| BC3+ | Denuded | RDG | 405 | UCD CRD-PAS | 1.80 + 2.05i | 10.78 (±3.11) | | 27 |
| BC3+ | Denuded | Mie | 532 | UCD CRD-PAS | 1.46 + 0.54i | 6.03 (±1.12) | 118/0.70 | 22 |
| BC3+ | Denuded | RDG | 532 | UCD CRD-PAS | 1.80 + 0.76i | 5.61 (±0.46) | | 22 |
| BC3+ | Denuded | Mie | 630 | CAPS PM$_{SSA}$ | 1.78+0.45i | 3.93 (±0.90) | 188/0.93 | 22 |
| BC3+ | Denuded | RDG | 630 | CAPS PM$_{SSA}$ | 1.80 + 0.55i | 3.51 (±1.55) | | 22 |

[%] Uncertainties in $MACs$ are 1$\sigma$ from the least $\chi^2$ fit. See Supplementary Figures S7-8.

[$] The $n$ values from the RDG method are non-unique. Therefore, uncertainty estimates from this work are not available. See text for details.

[@] VED and the size parameter (x = $\pi d_p/\lambda$) where the peak $MAC$ occurs

[*] There are many degenerate RI combinations that give similar quality fit to RDG theory. Thus, a value of 1.80 was chosen for the effective real refractive index.

[#] In BC2 the ethylene flat-burner flame was sampled 20.3 cm between the burner surface and the sampling inlet and during BC3 the flame was sampled 2" between the burner surface and the sampling inlet.





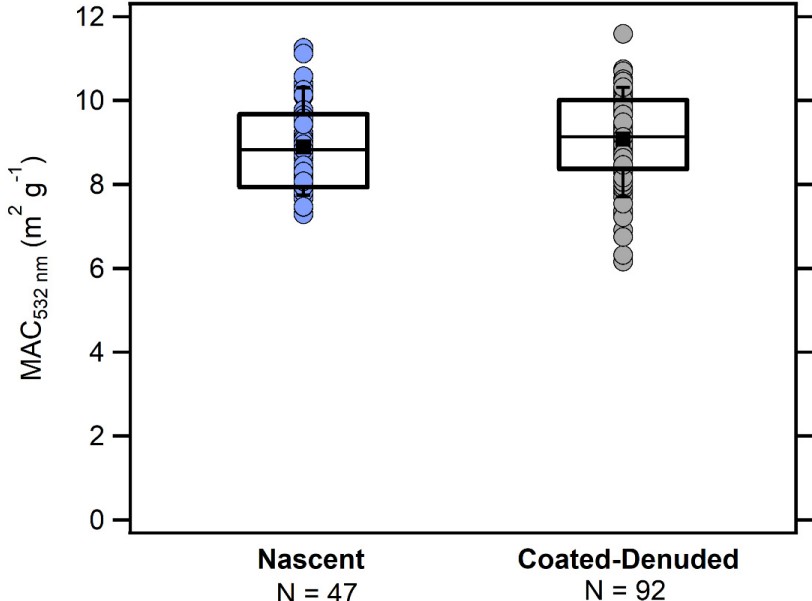

**Figure 1.** A comparison of methane diffusion-flame generated nascent (or nascent denuded; blue circles) and coated-denuded (grey circles) mass absorption coefficients (*MACs*) at $\lambda = 532$ nm for $x > 0.90$ for BC3, BC3+, and BC4 experiments. Note that conditioning (and therefore the shape) of the soot does not affect the observed *MACs*. The box and whisker plots show the mean (■), median (-), lower and upper quartile (boxes) and 9th and 91st percentile (whisker).





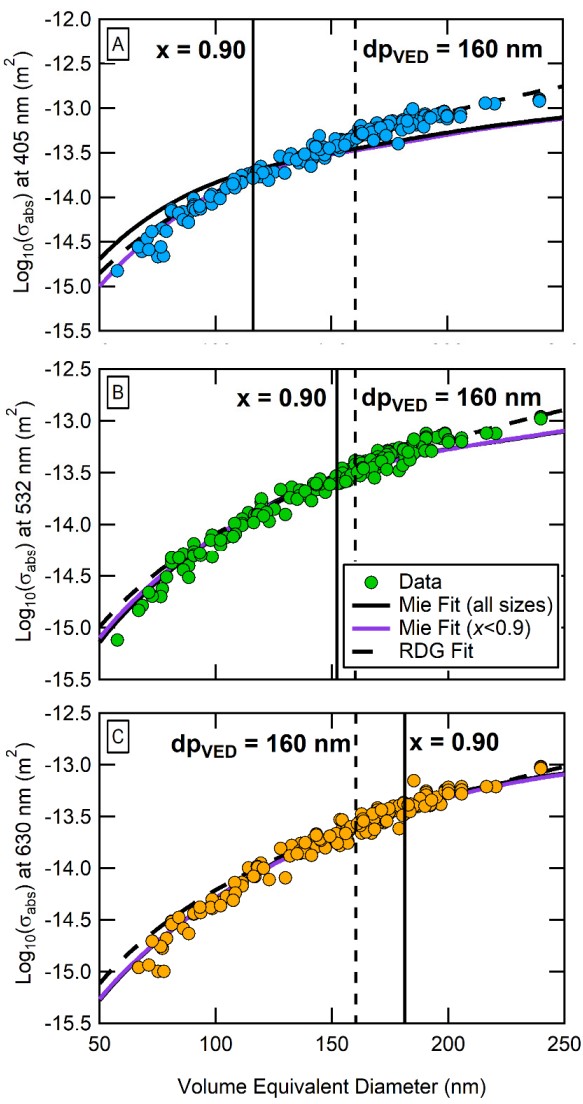

**Figure 2.** Observed $\sigma_{abs}$ versus $d_{p,VED}$ methane soot data from BC3,BC3+, and BC4. Panels A and B are $\lambda = 405$ nm and 532 nm data, respectively, from the UCD CRD-PAS and panel C is $\lambda = 630$ nm data from the CAPS PM$_{SSA}$. The nascent and coated-denuded data have been combined since there is no significant difference in the absorption cross-sections and $MAC$s between the two datasets. Note the inability of Mie Theory to reproduce the observed $\sigma_{abs,405nm}$ for all sizes. Vertical solid lines indicating $x = 0.9$, where observations deviate from Mie theory, are provided for reference. In addition, vertical dashed lines indicate $d_{p,VED} = 160$ nm, above which soot maturity is approximately constant.





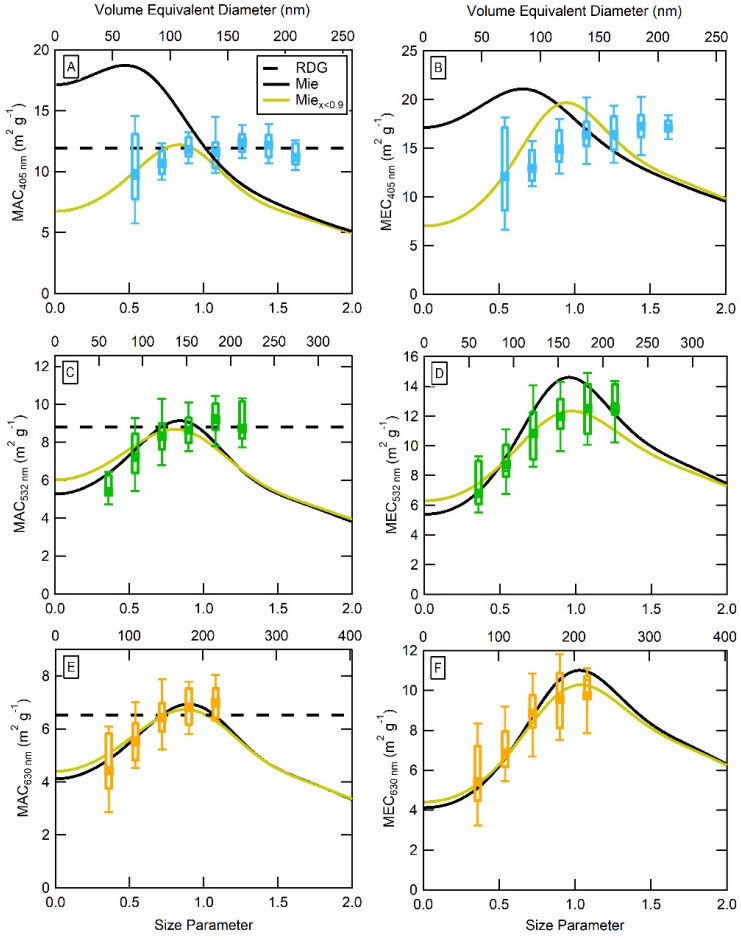

**Figure 3.** Box plots of *MAC*s and *MEC*s as a function of size parameter, with $\Delta x = 0.18$. The volume equivalent diameters are provided at the top of the plots for reference. Also shown are Mie theory curves for all particles (black solid lines), Mie theory curves for $x < 0.90$ (gold line), and RDG curves (dashed line) calculated from the RI values in Table 1. Panels A and B are $\lambda = 405$ nm data from the UCD CRD-PAS, panels C and D are $\lambda = 532$ nm data from the UCD CRD-PAS, and panels E and F are $\lambda = 630$ nm data from the CAPS PM$_{SSA}$. The poor match between the calculated Mie theory curves at $\lambda = 405$ nm reflects the difficulties in fitting spherical particle Mie theory to the observed $\sigma_{abs,405nm}$ over the entire size range. Although the same particle sizes were sampled over all wavelengths, the size parameters sampled at each wavelength are different. Therefore, there are different numbers of boxes for each wavelength. Note that this figure is directly related to Figure 2, the difference being that the y-axis values in Figure 2 (cross-sections) have been divided by the per particle mass to give the *MAC* or *MEC*. Points in each bin range from N=10 at $x_{1.62}$ to N = 81 at $x_{1.26}$ for $\lambda = 405$ nm, N = 5 at $x_{0.36}$ to N = 82 at $x_{1.08}$ at $\lambda = 532$ nm, and N=8 at $x_{1.08}$ to N=67 at $x_{0.9}$ at $\lambda = 630$ nm.



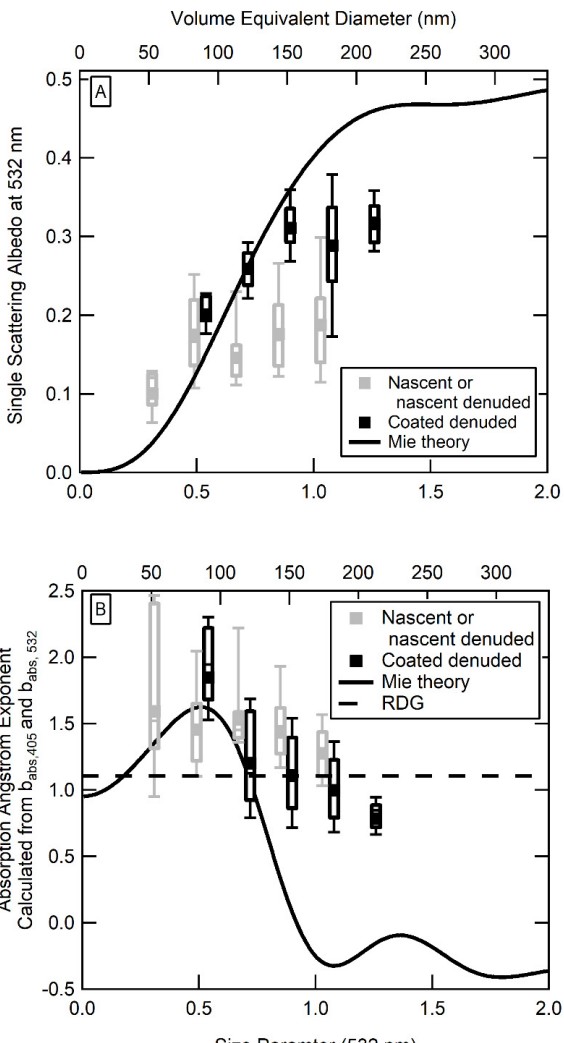

**Figure 4.** Box and whisker plots of $\lambda = 532$ nm (A) single scattering albedo (*SSA*) and (B) absorption Ångström exponent (*AAE*) as a function of volume equivalent diameter produced from the methane diffusion flame. The black points are coated-denuded data (coating material is evaporated following coating with DOS or sulfuric acid) and are potentially collapsed due to coating and the grey points are nascent or nascent denuded data. The black lines are the *SSA* or *AAE* predicted by spherical particle Mie theory and the dashed black line in panel B is *AAE* predicted from the RDG approximation using the effective refractive indices listed in Table 1. The nascent or nascent denuded boxes (black) are shifted by X = 0.05. Points in each bin range from N=6 at $x_{0.36}$ to N = 26 at $x_{1.08}$ for nascent (or nascent denuded) and N=5 at $x_{0.54}$ to N = 9 at $x_{1.26}$ for coated-denuded points.



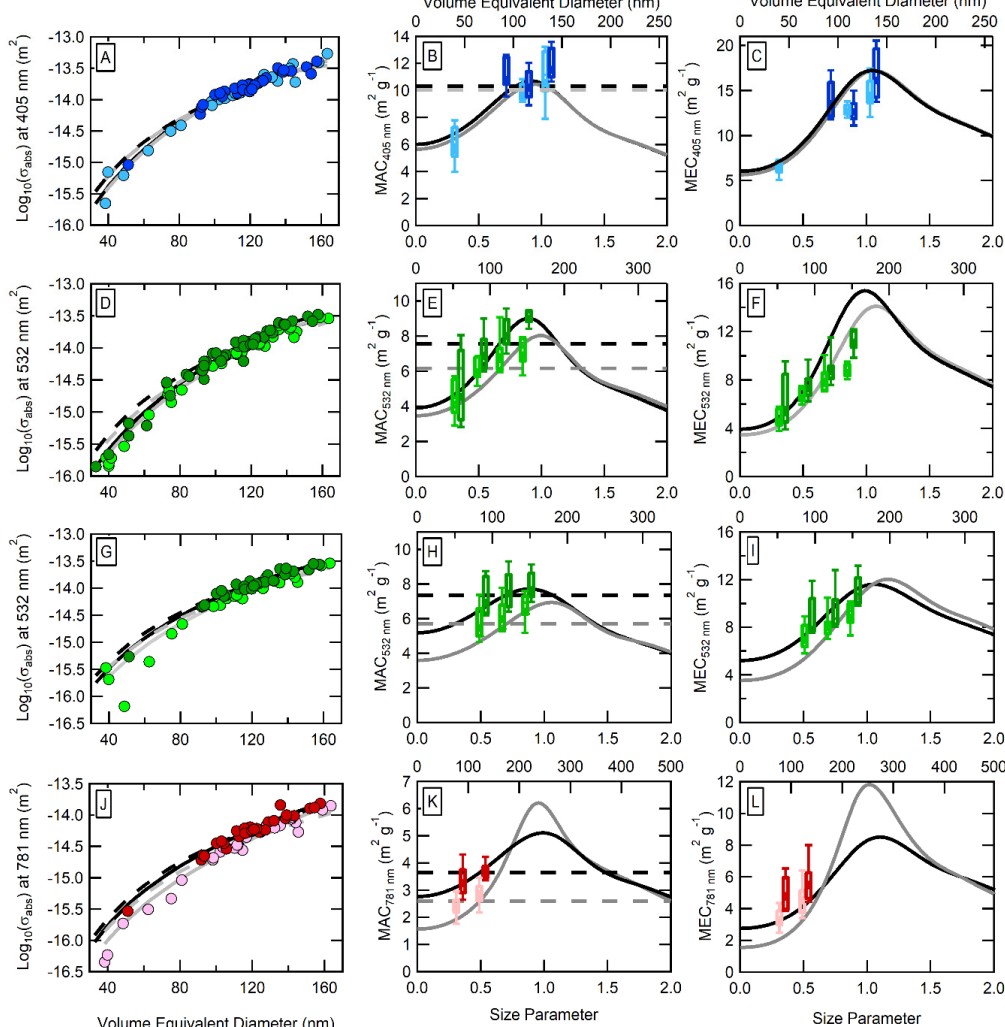

**Figure 5.** Measured absorption cross-sections vs. volume equivalent diameter (panels in the first column) and mass absorption and extinction coefficients (*MAC*s and *MEC*s; panels in the second and third column, respectively) for ethylene soot sampled 20.3 cm from burner surface of the McKenna flame as a function of size parameter (x = $\pi d_p/\lambda$) (bottom axes) and volume equivalent diameter (top axes). for BC2. Panels A, B, and C show PASS-3 data at λ = 405 nm, panels D, E, F show NOAA PAS data at λ = 532 nm, panels G, H, I show PASS-3 data at λ = 532 nm, and panels J, K, and L show PASS-3 data at λ = 781 nm. The dashed black and grey lines are the RDG fits to denuded and nascent data, respectively, and the solid black and grey lines are Mie fits to denuded and nascent data, respectively.


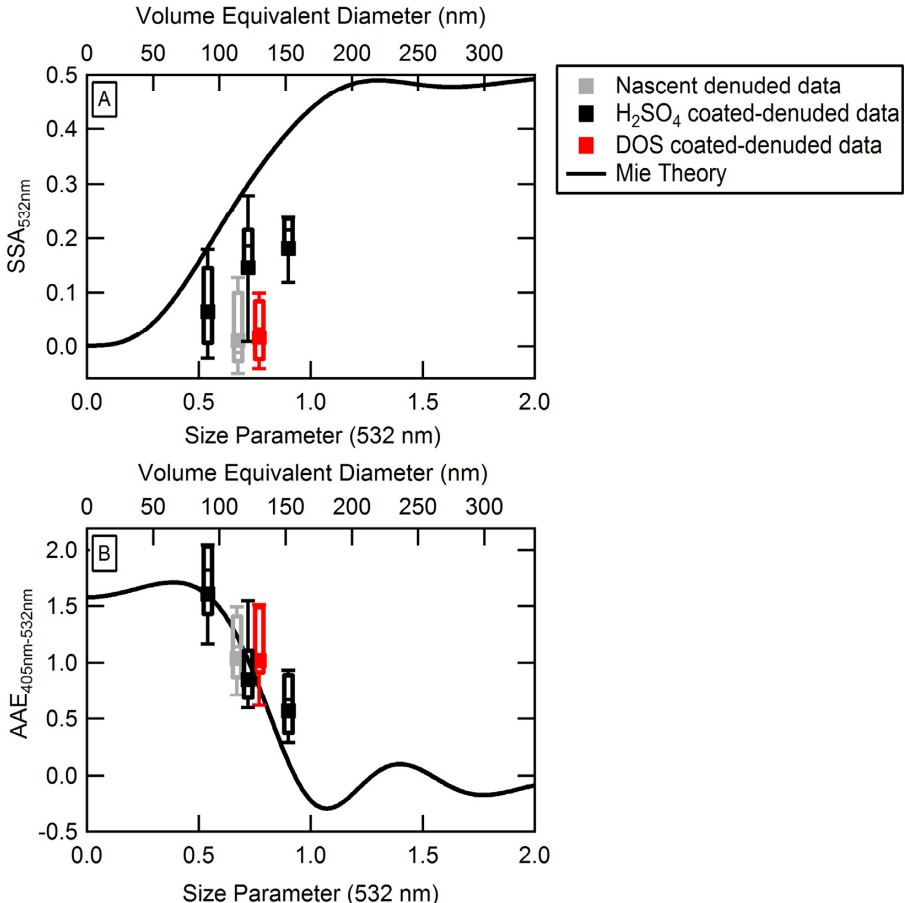

**Figure 6.** Box and whisker plots of (A) single scattering albedo (*SSA*) and (B) absorption Ångström exponent (*AAE*) as a function of size parameter and volume equivalent diameter produced from the ethylene flame during BC2. The *SSA* was calculated using data from the NOAA PAS and the *AAE* was calculated using data from the $\lambda = 405$ nm PASS-3 and the $\lambda = 532$ nm NOAA PAS data. The black points are coated-denuded data (coating material is evaporated following coating with DOS or sulfuric acid) and are potentially collapsed due to coating and the gray points are nascent denuded points (evaporation of intrinsic organic matter produced from the ethylene flame). The black lines are the *SSA*s predicted using spherical particle Mie theory using the effective complex refractive index retrieved from fitting the $\sigma_{abs,532nm}$ from the NOAA PAS. Here points are binned with a constant $\Delta x = 0.18$ for the purpose of comparison between the different particle treatments. N = 6 for nascent or nascent denuded points, N=5 for DOS coated-denuded points, and $N_{x=0.54} = 11$, $N_{x=0.72} = 8$, and $N_{x=0.9} = 5$ for $H_2SO_4$ coated-denuded points.