# Peer review of "Measurement and modeling of the multi-wavelength optical properties of uncoated flame-generated soot"

_Atmospheric Chemistry and Physics, 2018_

## Referee Comment (RC1) · Anonymous Referee #1 · 11 May 2018

Review of 'Measurement and modeling of the multi-wavelength optical properties of uncoated flame-generated soot'

Overall impression

The manuscript 'Measurement and modeling of the multi-wavelength optical properties of uncoated flame-generated soot' by Forestieri et al. presents a detailed analysis of laboratory studies of black carbon particles to rationalize a more appropriate (Rayleigh-Debye-Gans approximation) that uses a constant mass absorption coefficient that is size independent, but wavelength specific treatment of uncoated BC particles within climate models. The authors present a clear question, clearly answer it and the findings

fall within the scope of ACP. The manuscript is technical, well-written, and provides a clear connection between laboratory experiments and conclusions. I have no major comments and recommend this manuscript for publication.

General comments

Significant time is devoted to placing these measurements in context, but there is frequently no suggestion as to why these measurements are different – when they are that is. I suggest providing the reader some guidance as to the authors view on some differences.

While these data support the conclusion presented in the abstract, it appears from Figure 3 that RDG theory over-predicts the MAC at smaller particle sizes when Mie Theory appears to do a better job, would a combination of these two separate approaches result in a better parameterization? It seems to be that the problem with black carbon absorption may simply be shifted from an underestimation with Mie Theory to and overestimation with RDG.

It's noted in the conclusions that future works will consider coatings, I look forward to seeing this work.

Specific comments

Pg 2, Line 6: consider defining Mie size parameter explicitly.

Pg 3, Line 19: 'fuller' seems out of place, perhaps 'more complete'?

Pg 4, Line 10: what is the associated uncertainty the fuel equivalence ratio?

Pg 8, Line 4: 'size-selected', mobility selected?

Pg 8, Line 4: Which soot, i.e. what lot from which company? There is a certain amount of variability?

Pg 8, Line 4: Was any coating analysis performed on the data collected from the SP2?

[Figure]

Pg 9, Line 18: perhaps citing the original paper (Park et al. 2004 I believe, there may have been one earlier that used the formulation shown in this manuscript) that brought up the Df,m mobility shape factor would be more appropriate than Cross et al. 2010.

Pg 12, Section 5.1.1: As I understand, the methane diffusion flame was used in BC3, BC3+, and BC4. Can you explain why the Df described here is different than that of the Df described in Bhandari et al., 2016? I would expect nascent soot particles to have a lower mass mobility exponent.

Pg 13, Line 27: It appears that Mie also overestimates at smaller particle sizes as well.

Pg 14, Line 6: Can you comment on why these RI values are larger than those used in current global climate models?

Pg 14, Line 8: Specify that this recommendation is for 550 nm.

Pg 14-15, Line various: considerable time is spent discussing the MAC values in context with other literature, but there is little comment as to why these differences may exist.

Pg 17, Line 16: I suggest adding this note "RI fitting was performed..." to the caption in Figure 3.

Pg 19, Line 1: Could information regarding the soot maturity be accessed using LEO-fitting analysis with the SP2? At least that would indicate coating thickness which would indicate atmospheric processing.

Figure 2: As presented the fits are hidden behind the data, please bring them to the front to allow the reader to clearly see them. Perhaps a bottom panel showing the % difference or some such metric indicating the deviation from model vs measurement below each plot to clearly identify differences between Mie and RDG could be helpful?

---

## Referee Comment (RC2) · Anonymous Referee #2 · 23 May 2018

The manuscript presents a thorough description of optical property measurements conducted during a series of experiments examining soot emissions from two different types of flames. It presents a detailed analysis of the merits of Mie and RDG approximations of uncoated soot optical properties, and explores implications for climate models. The experiments and measurements are of a high quality and provide an extremely useful collection of data for interpretation of similar measurements performed for other BC emission sources. They raise important questions regarding treatment of BC in climate models, and I look forward to their future work related to coated BC properties.

[Figure]

I recommend its publication in ACP with only a few minor corrections, listed below.

The reference in the introduction giving an upper end estimate of potential BC forcing is now 10 years old, and this section would benefit form including one or two more recent estimates, though I understand the authors are pointing out an extreme case.

Page 8, line 4: please be specific as to what "size" is referring to here...mobility diameter or mass.

Page 8, line 22: stating truncation angles for the CAPS-SSA here would be helpful.

Page 9 - i believe the equations for MAC should have units of area, not inverse Mm.

———————————————————

---

## Referee Comment (RC3) · Anonymous Referee #3 · 26 May 2018

This is a well written article and present the solid absorption result using the well con-strained soot source and the result is straightforward for models to pick up. but it would even more benefit the community if addressing the following points:

-how could flame soot represent the ambient soot, in terms of refractive index and particle morphology? Then how could be suggested these results could be widely used in the model?

-if we have a different source of BC, for example the biomass burning, how could we guarantee the RI still the same?

-It would be better to show the mass distribution of DMA-selected particles at different

cases (to indicate the width of the distribution), as the single particle mass is crucial for the following analysis.

-how have you proven the TD 5secs soot is nascent or no re-condensation down the pipe? Maybe showing some mass spectra to prove these are all refractory BC will be useful. And this also concerns the coated and then denuded soot.

-It was mentioned you have used three PASS instruments, how were they compared with each other? better to show in a plot maybe.

-Fig. S4, could we change the colour scale a bit show the minima of X2.

-there is no label for Fig. S8.

-one important information is how the size parameter could relate to the volume equivalent diameter. For a general practice, could we assume >160nm BC will have a MAC using RDG approach, and how this VED will depend on the wavelength.

---

## Short Comment (SC1) · 31 May 2018

This manuscript is well written. I have a minor suggestion. Page 3, Lines 5–8: For the authors' information, another new particle-optics method (geometric-optics surface-wave approach) has been recently developed to calculate optical properties of BC with complex particle structures (He et al., 2015, 2016), which also accounts for interactions between spherules and could be included as a reference here.

References

He, C., et al.: Variation of the radiative properties during black carbon aging: theo-

retical and experimental intercomparison, Atmos. Chem. Phys., 15, 11967-11980, doi:10.5194/acp-15-11967-2015, 2015.

He, C., et al.: Intercomparison of the GOS approach, superposition T-matrix method, and laboratory measurements for black carbon optical properties during aging, J. Quant. Spectrosc. Radiat. Transf., 184, 287–296, doi:10.1016/j.jqsrt.2016.08.004, 2016.
* * *

---

## Author Comment (AC1) · 20 Jul 2018

We thank the reviewers for their thoughtful comments, and address each point below. We note changes that have been made to the manuscript. Original reviewer comments are in **black** and our responses in blue. New text added is *italicized*.

**Response to Reviewer #1**

Overall impression
The manuscript 'Measurement and modeling of the multi-wavelength optical properties of uncoated flame-generated soot' by Forestieri et al. presents a detailed analysis of laboratory studies of black carbon particles to rationalize a more appropriate (Rayleigh- Debye-Gans approximation) that uses a constant mass absorption coefficient that is size independent, but wavelength specific treatment of uncoated BC particles within climate models. The authors present a clear question, clearly answer it and the findings fall within the scope of ACP. The manuscript is technical, well-written, and provides a clear connection between laboratory experiments and conclusions. I have no major comments and recommend this manuscript for publication.

We thank the reviewer for the thoughtful review.

Significant time is devoted to placing these measurements in context, but there is frequently no suggestion as to why these measurements are different – when they are that is. I suggest providing the reader some guidance as to the authors view on some differences.

This is a good point. We have updated the discussion to provide some guidance. One likely reason for differences—when they exist—is that soot of varying maturity was accessed in a given study. As is suggested by Fig. 3 and the discussion in Section 5.1.3 (and in the response to the next comment), the MAC likely increases with maturation. Thus, in cases where literature MAC values are on the lower side, it could reflect sampling of less mature soot. Of course, differences could simply result from experimental uncertainty and historical challenges in measuring both BC concentrations and absorption accurately.

While these data support the conclusion presented in the abstract, it appears from Figure 3 that RDG theory over-predicts the MAC at smaller particle sizes when Mie Theory appears to do a better job, would a combination of these two separate approaches result in a better parameterization? It seems to be that the problem with black carbon absorption may simply be shifted from an underestimation with Mie Theory to and overestimation with RDG.

While this is certainly possible, we believe that the more likely reason for the overestimate at small sizes results from challenges with generating particles with constant composition ("maturity") at all sizes using the flame generators here. This is directly related to the discussion of soot "maturity" in section 5.1.3. While we do not have definitive proof, we believe that had we been able to sample soot of a constant maturity at all sizes we would have observed a constant (size-independent) MAC. Soot maturity has been studied for flames and fuels such as that used here by others (Johansson et al., 2017;Leschowski et al., 2015;López-Yglesias et al., 2014). The work by Johansson et al. (2017) suggests that the absorption cross-section (and thus the MAC) increases with maturity, consistent with our

observations. Additionally, they note that the wavelength dependence of absorption is steeper (higher AAE) for less mature soot. This is consistent with our observations in Fig. 4, where there is, perhaps, a slight downward trend in the AAE with particle size.

An important question is, of course, what is the maturity of soot produced from diesel engines and biomass combustion, does it vary with size, and how does it relate to the soot produced here? As we noted, "flame-generated BC particles have been shown to be a suitable proxy for atmospheric BC particles, both in terms of chemical bonding and structural properties (Slowik et al., 2007;Hopkins et al., 2007)." However, the question of size-dependent maturity for diesel and biomass burning particles is unresolved. Yon et al. (2011)investigated soot produced from combustion of diesel fuel, albeit produced from a flame generator and not an engine, but did not look at the size dependence. Their results do, however, suggest that the soot from diesel soot can be considered reasonably mature.

It's noted in the conclusions that future works will consider coatings, I look forward to seeing this work.

Indeed. We hope to have this completed soon.

Specific comments

Pg 2, Line 6: consider defining Mie size parameter explicitly.

We have added an explicit definition here, and on page 12. "(*The dimensionless size parameter $x = \pi d_p/\lambda$, where $d_p$ is particle diameter and l is wavelength*.)"

Pg 3, Line 19: 'fuller' seems out of place, perhaps 'more complete'?

We have updated to "*more thorough.*"

Pg 4, Line 10: what is the associated uncertainty the fuel equivalence ratio?

We have updated this as follows:

"During BC3, BC3+, and BC4, most experiments were conducted using particles produced from an inverted co-flow diffusion flame operating on methane with a sheath flow mixture of $O_2$ and $N_2$ (Stipe et al., 2005) with a net fuel equivalence ratio, *$\phi$, = 0.7 ± 0.07; because the diffusion flame entrains sheath oxygen into the methane-rich center flow a range of $\phi$ values are accessed including regions where the local $\phi$ may be greater than 1.*"

Pg 8, Line 4: 'size-selected', mobility selected?

Yes. We have clarified to now state "*DMA size-selected.*"

Pg 8, Line 4: Which soot, i.e. what lot from which company? There is a certain amount of variability?

We now report that we used Alfa Aesar Lot L18U002.

Pg 8, Line 4: Was any coating analysis performed on the data collected from the SP2?

No, we did not perform coating analysis using the SP2 for this study. The laboratory environment allowed us to constrain the amount of coating using the CPMA.

Pg 9, Line 18: perhaps citing the original paper (Park et al. 2004 I believe, there may have been one earlier that used the formulation shown in this manuscript) that brought up the Df,m mobility shape factor would be more appropriate than Cross et al. 2010.

We now cite Park et al. (2003), who report this relationship.

Pg 12, Section 5.1.1: As I understand, the methane diffusion flame was used in BC3, BC3+, and BC4. Can you explain why the Df described here is different than that of the Df described in Bhandari et al., 2016? I would expect nascent soot particles to have a lower mass mobility exponent.

First, we have updated this reference to Bhandari et al. (2017). There is fundamental difference between the fractal dimension (what Bhandari et al measure) and the mass-mobility exponent (what is measured in this study). The difference between the two is described in detail by Sorensen (2011). The fractal dimension of typical soot is on the order of ~1.8; the mass-mobility exponent of typical uncoated soot is on the order of ~2.5 (e.g. Figure 3 in Olfert et al. (2017)). Thus we fully expect the fractal dimension measured by Bhandari (2017) to be lower than the mass-mobility exponent measured here.

To the reviewer's question, the difference likely results from different methods of determination. Bhandari et al. (2017) determined the Df (fractal dimension) values from analysis of SEM images and by fitting a line to the number of spherules comprising a particle versus dimensions of the particle (specifically $\sqrt{(L \cdot W)/d_p}$ where $L$ and $W$ are the length and width of the overall particle and $d_p$ is the spherule diameter). In contrast, we determined Df,m (mass mobility exponent) from plots of per-particle mass versus mobility diameter.

We have added the following to Section 4.1: "(The mass mobility exponent, $D_{f,m}$, differs from the fractal dimension, $D_f$, as discussed by Sorensen (2011).)"

Pg 13, Line 27: It appears that Mie also overestimates at smaller particle sizes as well.

We have updated this sentence to clarify: "When the fits are restricted to $x < 0.9$, a reasonable fit using Mie theory is obtained at all wavelengths over this size range, *although there is perhaps a small overestimate at the smallest sizes*."

Pg 14, Line 6: Can you comment on why these RI values are larger than those used in current global climate models?

In short, because many models have adopted the values of Bond and Bergstrom (2006), and their suggested RI yields MAC values at many sizes lower than is supported by observations. We have added the following for clarification: "For reference, using $RI = 1.95 – 0.79i$ the $MAC$ at 532 nm calculated for BC in the small particle limit (assuming a material density of 1.8 g/cm$^3$) is only 5.1 m$^2$ g$^{-1}$, but peaks at 7.5 m$^2$ g$^{-1}$ around $d_{p,VED} = 150$ nm."

For additional consideration, the value suggested by Bond and Bergstrom (2006) was arrived at after consideration of a variety of literature measurements of soots of various types, and with measurements made in many different ways. They determined a value of $1.95 – 0.79i$ by matching, approximately, a model that accounts for varying "void fractions" with a suite of the observations, and finding where this intersects with an estimate of how the refractive index varies with extent of "graphitization." They ultimately state that "The value 1.95–0.79i merely provides agreement with many of the measurements."

Pg 14, Line 8: Specify that this recommendation is for 550 nm.

It is our understanding that the RI reported by Bond and Bergstrom (2006) is not for a specific wavelength, but is meant to be general. The references used to derive the RI do not all use 550 nm. The recommended MAC value is, however, wavelength-specific.

Pg 14-15, Line various: considerable time is spent discussing the MAC values in context with other literature, but there is little comment as to why these differences may exist.

We think the most likely reason for differences is differences in soot maturity. We have added a sentence to this effect. "One key reason that differences may exist between studies is that the BC particles sampled had differing maturity. Soot maturity refers to the extent to which the BC has a more disordered internal structure with high hydrogen content (low maturity) versus a more ordered, graphite-like structure with low hydrogen content (high maturity) (Johansson et al., 2017). The absorption cross-section for BC likely increases with increasing soot maturity (López-Yglesias et al., 2014)."

Pg 17, Line 16: I suggest adding this note "RI fitting was performed. . ." to the caption    in Figure 3.

We have updated Fig. 3 caption.

Pg 19, Line 1: Could information regarding the soot maturity be accessed using LEO- fitting analysis with the SP2? At least that would indicate coating thickness which would indicate atmospheric processing.

To clarify, the concept of "soot maturity" is not related to how coated (or not) a particle is, but to the properties of the BC material (see above). As such, LEO-fitting would not be especially helpful.

Figure 2: As presented the fits are hidden behind the data, please bring them to the  front to allow the reader to clearly see them. Perhaps a bottom panel showing the % difference or some such metric

indicating the deviation from model vs measurement below each plot to clearly identify differences between Mie and RDG could be helpful?

We have updated Fig. 2 to move the fit lines to the front. We have not added additional % difference plots because Fig. 3 shows the transformation of the data in Fig. 2 to MAC space, and we think that this further helps to illustrate the differences.

**Response to Reviewer #2**

The manuscript presents a thorough description of optical property measurements con- ducted during a series of experiments examining soot emissions from two different    types of flames. It presents a detailed analysis of the merits of Mie and RDG ap- proximations of uncoated soot optical properties, and explores implications for climate models. The experiments and measurements are of a high quality and provide an ex- tremely useful collection of data for interpretation of similar measurements performed for  other BC emission sources.  They raise important questions regarding treatment      of BC in climate models, and I look forward to their future work related to coated BC properties

I recommend its publication in ACP with only a few minor corrections, listed below.

The reference in the introduction giving an upper end estimate of potential BC forcing is now 10 years old, and this section would benefit form including one or two more recent estimates, though I understand the authors are pointing out an extreme case.

We have updated this as follows:

> "The exact magnitude of the climate impacts of BC remain uncertain. One estimate puts top-of-the-atmosphere direct forcing by BC as high as 0.9 W $m^{-2}$, which is comparable in magnitude to that of $CO_2$ (Ramanathan and Carmichael, 2008). Other more recent assessments yield 0.71 W $m^{-2}$ with 90% uncertainty bounds of 0.08 to 1.27 W $m^{-2}$ (Bond et al., 2013) or 0.61 [+0.16 to +1.40] W $m^{-2}$ (Wang et al., 2016), while the IPCC suggests a value of 0.40 [+0.05 to +0.80] W $m^{-2}$ (Boucher et al., 2013)."

Page 8, line 4: please be specific as to what "size" is referring to here...mobility diam- eter or mass.

We now state "DMA size selected" to clarify.

Page 8, line 22: stating truncation angles for the CAPS-SSA here would be helpful.

Rather than giving the truncation angles, we now state the magnitude of the correction based on (Onasch et al., 2015b).

> "The CAPS PM$_{SSA}$ measures $b_{sca}$ using an integrating nephelometer, corrected for the finite viewing angle of the detector, i.e. truncation correction (Onasch et al., 2015b). *The truncation*

*correction at 630 nm was determined to be <1% at 630 nm for particles smaller than 300 nm, increasing to <5% for particles smaller than 800 nm."*

Page 9 - i believe the equations for MAC should have units of area, not inverse Mm

We think the reviewer meant cross-sections, not MAC. We have updated the manuscript.

**Response to Reviewer #3**

This is a well written article and present the solid absorption result using the well con- strained soot source and the result is straightforward for models to pick up. but it would even more benefit the community if addressing the following points:

-how could flame soot represent the ambient soot, in terms of refractive index and particle morphology? Then how could be suggested these results could be widely used in the model?

-if we have a different source of BC, for example the biomass burning, how could we guarantee the RI still the same?

We address the above two points together. While we do not have direct evidence that the BC produced from our flames is guaranteed to have the same RI as ambient soot, we point to the references of Hopkins et al. (2007) and Slowik et al. (2007) as support for the lab soot and ambient soot having similar properties. As we previously stated:

"Although atmospheric BC particles are predominately generated through combustion of fossil fuels or through biomass burning (Bond et al., 2013), flame-generated BC particles have been shown to be a suitable proxy for atmospheric BC particles, both in terms of chemical bonding and structural properties (Slowik et al., 2007;Hopkins et al., 2007)."

We have expanded this discussion (in Section 5.1.3) to include additional details:

"For example, Hopkins et al. (2007) find that the $sp^2$ content of ethylene and methane flame soot are similar to diesel soot (63%, 60%, and 56%, respectively), and have similar aromatic content. There is also a reasonable similarity between SP-AMS mass spectra of flame soot and soot particles in diesel exhaust or smoke from biomass burning (Onasch et al., 2015a)."

-It would be better to show the mass distribution of DMA-selected particles at different cases (to indicate the width of the distribution), as the single particle mass is crucial for the following analysis.

We now report typical geometric standard deviations ($\sigma_{g,CPMA}$) in the measured per-particle mass from the CPMA. For "forward-coating" experiments, the typical $\sigma_{g,CPMA}$ was 1.3. For "reverse-coating" experiments, the $\sigma_{g,CPMA}$ were generally larger, ranging from around 1.3 to 2. Based on the reviewers concern, we have examined whether there is any dependence of the measured MAC values on the

$\sigma_{g,CPMA}$. Using data from BC4 as an example, we find no significant dependence of the measured MAC values on $\sigma_{g,CPMA}$, with a linear fit between MAC and $\sigma_{g,CPMA}$ giving an $r^2 < 0.01$ and a slope at 532 nm (for example) of -0.08 +/- 0.7. We now state in Section 3.2 that:

*"The typical geometric standard deviation of the per-particle mass distributions for forward-coating experiments was 1.3, while for reverse-coating experiments it ranged from around 1.3 to 2."*

And in section 5.1.1 we now state that:

*"For the reverse-coating experiments, which gave broader BC per-particle mass distributions compared to forward-coating distributions, we found no dependence of the derived MAC values on the distribution width."*

-how have you proven the TD 5secs soot is nascent or no re-condensation down the pipe? Maybe showing some mass spectra to prove these are all refractory BC will be useful. And this also concerns the coated and then denuded soot.

The removal of coating material was demonstrated for BC2 by Cross et al. (2010) for these coating materials, and is further supported by the literature. While we could add a few spectra if deemed really necessary, we think that it is sufficient to add the following statement in Section 3.2: *"Literature results indicate that thermodenuding of particles coated with or composed of these materials leads to essentially complete evaporation of the non-refractory material (Cappa and Wilson, 2011;Huffman et al., 2008), confirmed by measurements from BC2 (Cross et al., 2010)."*

-It was mentioned you have used three PASS instruments, how were they compared with each other? better to show in a plot maybe.

This was only the case for the BC2 study, and is already discussed in Cross et al. (2010). Rather than repeat this analysis, we now point the reader to this reference.

-Fig. S4, could we change the colour scale a bit show the minima of X2.

For the RDG fits, the solution surface is very flat and there are many values that yield equivalently good fits to the observations. We show below Fig. S4A with the original scale (left) and an extremely narrowed scale (right). Even with changing the scale, the minima in the surface is not evident. The key point is that there is not a single solution for the RDG fits. As we stated in the main text:

*"In contrast, fits performed using the RDG approximation do not give a unique set of $m$ and $k$, but instead a band of [$m,k$] pairs that describe the data equally well (Figure S4). Since RDG fits are non-unique, optimal $k$ values are reported at all wavelengths for a fixed value of $m = 1.80$."*

As such, we have not updated the figures. We have, however, updated the caption to indicate that the crosses for the RDG fits were selected where $m = 1.80$.

[Figure]

-there is no label for Fig. S8.

Perhaps there was a pdf rendering problem? We downloaded the supplement and can see the figure caption for Fig. S8.

-one important information is how the size parameter could relate to the volume equivalent diameter. For a general practice, could we assume >160nm BC will have a MAC using RDG approach, and how this VED will depend on the wavelength.

The size parameter relates to the volume equivalent diameter as: $x = \pi d_{p,VED}/\lambda$. We now give this definition explicitly. Yes, for larger particles it can be assumed that the MAC of BC can be represented using an RDG approach. This is equivalent to assuming a constant MAC, as we conclude in the last sentence of our abstract.

**References**

Bhandari, J., China, S., Onasch, T., Wolff, L., Lambe, A., Davidovits, P., Cross, E., Ahern, A., Olfert, J., Dubey, M., and Mazzoleni, C.: Effect of Thermodenuding on the Structure of Nascent Flame Soot Aggregates, Atmosphere, 8, 166, 10.3390/atmos8090166, 2017.

Bond, T. C., and Bergstrom, R. W.: Light Absorption by Carbonaceous Particles: An Investigative Review, Aerosol Science and Technology, 40, 27-67, 10.1080/02786820500421521, 2006.

Bond, T. C., Doherty, S. J., Fahey, D. W., Forster, P. M., Berntsen, T., DeAngelo, B. J., Flanner, M. G., Ghan, S., Kärcher, B., Koch, D., Kinne, S., Kondo, Y., Quinn, P. K., Sarofim, M. C., Schultz, M. G., Schulz, M., Venkataraman, C., Zhang, H., Zhang, S., Bellouin, N., Guttikunda, S. K., Hopke, P. K., Jacobson, M. Z., Kaiser, J. W., Klimont, Z., Lohmann, U., Schwarz, J. P., Shindell, D., Storelvmo, T., Warren, S. G., and Zender, C. S.: Bounding the role of black carbon in the climate system: A scientific assessment, Journal of Geophysical Research: Atmospheres, 118, 5380-5552, 10.1002/jgrd.50171, 2013.

Boucher, O., Randall, D., Artaxo, P., Bretherton, C., Feingold, G., Forster, P., Kerminen, V.-M., Kondo, Y., Liao, H., Lohmann, U., Rasch, P., Satheesh, S. K., Sherwood, S., Stevens, B., and Zhang, X. Y.: Clouds and Aerosols, in: Climate Change 2013: The Physical Science Basis. Contribution of Working Group I to the Fifth Assessment Report of the Intergovernmental Panel on Climate Change, edited by: Stocker, T. F., Qin, D., Plattner, G.-K., Tignor, M., Allen, S. K., Boschung, J., Nauels, A., Xia, Y., Bex, V., and Midgley, P. M., Cambridge University Press, Cambridge, United Kingdom and New York, NY, USA, 571-658, 2013.

Cappa, C. D., and Wilson, K. R.: Evolution of organic aerosol mass spectra upon heating: implications for OA phase and partitioning behavior, Atmos. Chem. Phys., 11, 1895-2011, 10.5194/acp-11-1895-2011, 2011.

Cross, E. S., Onasch, T. B., Ahern, A., Wrobel, W., Slowik, J. G., Olfert, J., Lack, D. A., Massoli, P., Cappa, C. D., Schwarz, J. P., Spackman, J. R., Fahey, D. W., Sedlacek, A., Trimborn, A., Jayne, J. T., Freedman, A., Williams, L. R., Ng, N. L., Mazzoleni, C., Dubey, M., Brem, B., Kok, G., Subramanian, R., Freitag, S., Clarke, A., Thornhill, D., Marr, L. C., Kolb, C. E., Worsnop, D. R., and Davidovits, P.: Soot Particle Studies—Instrument Inter-Comparison—Project Overview, Aerosol Science and Technology, 44, 592-611, 10.1080/02786826.2010.482113, 2010.

Hopkins, R. J., Tivanski, A. V., Marten, B. D., and Gilles, M. K.: Chemical bonding and structure of black carbon reference materials and individual carbonaceous atmospheric aerosols, Journal of Aerosol Science, 38, 573-591, 10.1016/j.jaerosci.2007.03.009, 2007.

Huffman, J. A., Ziemann, P. J., Jayne, J. T., Worsnop, D. R., and Jimenez, J. L.: Development and characterization of a fast-stepping/scanning thermodenuder for chemically-resolved aerosol volatility measurements, Aerosol Sci. Technol., 42, 395-407, 10.1080/02786820802104981, 2008.

Johansson, K. O., El Gabaly, F., Schrader, P. E., Campbell, M. F., and Michelsen, H. A.: Evolution of maturity levels of the particle surface and bulk during soot growth and oxidation in a flame, Aerosol Science and Technology, 51, 1333-1344, 10.1080/02786826.2017.1355047, 2017.

Leschowski, M., Thomson, K. A., Snelling, D. R., Schulz, C., and Smallwood, G. J.: Combination of LII and extinction measurements for determination of soot volume fraction and estimation of soot maturity in non-premixed laminar flames, Applied Physics B, 119, 685-696, 10.1007/s00340-015-6092-2, 2015.

López-Yglesias, X., Schrader, P. E., and Michelsen, H. A.: Soot maturity and absorption cross sections, Journal of Aerosol Science, 75, 43-64, 10.1016/j.jaerosci.2014.04.011, 2014.

Olfert, J. S., Dickau, M., Momenimovahed, A., Saffaripour, M., Thomson, K., Smallwood, G., Stettler, M. E. J., Boies, A., Sevcenco, Y., Crayford, A., and Johnson, M.: Effective density and volatility of particles sampled from a helicopter gas turbine engine, Aerosol Science and Technology, 51, 704-714, 10.1080/02786826.2017.1292346, 2017.

Onasch, T. B., Fortner, E. C., Trimborn, A. M., Lambe, A. T., Tiwari, A. J., Marr, L. C., Corbin, J. C., Mensah, A. A., Williams, L. R., Davidovits, P., and Worsnop, D. R.: Investigations of SP-AMS Carbon Ion Distributions as a Function of Refractory Black Carbon Particle Type, Aerosol Science and Technology, 49, 409-422, 10.1080/02786826.2015.1039959, 2015a.

Onasch, T. B., Massoli, P., Kebabian, P. L., Hills, F. B., Bacon, F. W., and Freedman, A.: Single Scattering Albedo Monitor for Airborne Particulates, Aerosol Science and Technology, 49, 267-279, 10.1080/02786826.2015.1022248, 2015b.

Park, K., Cao, F., Kittelson, D. B., and McMurry, P. H.: Relationship between Particle Mass and Mobility for Diesel Exhaust Particles, Environmental Science & Technology, 37, 577-583, 10.1021/es025960v, 2003.

Ramanathan, V., and Carmichael, G.: Global and regional climate changes due to black carbon, Nature Geosci, 1, 221-227, 10.1038/ngeo156, 2008.

Slowik, J. G., Cross, E. S., Han, J.-H., Davidovits, P., Onasch, T. B., Jayne, J. T., WilliamS, L. R., Canagaratna, M. R., Worsnop, D. R., Chakrabarty, R. K., Moosmueller, H., Arnott, W. P., Schwarz, J. P., Gao, R.-S., Fahey, D. W., Kok, G. L., and Petzold, A.: An inter-comparison of instruments measuring black carbon content of soot particles, Aerosol Science and Technology, 41, 295-314, 10.1080/02786820701197078, 2007.

Sorensen, C. M.: The Mobility of Fractal Aggregates: A Review, Aerosol Science and Technology, 45, 765-779, 10.1080/02786826.2011.560909, 2011.

Stipe, C. B., Higgins, B. S., Lucas, D., Koshland, C. P., and Sawyer, R. F.: Inverted co-flow diffusion flame for producing soot, Review of Scientific Instruments, 76, 10.1063/1.1851492, 2005.

Wang, R., Balkanski, Y., Boucher, O., Ciais, P., Schuster, G. L., Chevallier, F., Samset, B. H., Liu, J., Piao, S., Valari, M., and Tao, S.: Estimation of global black carbon direct radiative forcing and its uncertainty constrained by observations, Journal of Geophysical Research: Atmospheres, 121, 5948-5971, doi:10.1002/2015JD024326, 2016.

Yon, J., Lemaire, R., Therssen, E., Desgroux, P., Coppalle, A., and Ren, K. F.: Examination of wavelength dependent soot optical properties of diesel and diesel/rapeseed methyl ester mixture by extinction spectra analysis and LII measurements, Applied Physics B, 104, 253-271, 10.1007/s00340-011-4416-4, 2011.

---

## Author Response (AR1)

We thank the reviewers for their thoughtful comments, and address each point below. We note changes that have been made to the manuscript. Original reviewer comments are in **black** and our responses in **blue**. New text added is *italicized*.

**Response to Reviewer #1**

**Overall** impression**

The manuscript 'Measurement and modeling of the multi-wavelength optical properties of uncoated flame-generated soot' by Forestieri et al. presents a detailed analysis of laboratory studies of black carbon particles to rationalize a more appropriate (Rayleigh- Debye-Gans approximation) that uses a constant mass absorption coefficient that is size independent, but wavelength specific treatment of uncoated BC particles within climate models. The authors present a clear question, clearly answer it and the findings fall within the scope of ACP. The manuscript is technical, well-written, and provides a clear connection between laboratory experiments and conclusions. I have no major comments and recommend this manuscript for publication.

**We thank the reviewer for the thoughtful review.**

Significant time is devoted to placing these measurements in context, but there is frequently no suggestion as to why these measurements are different – when they are that is. I suggest providing the reader some guidance as to the authors view on some differences.

This is a good point. We have updated the discussion to provide some guidance. One likely reason for differences—when they exist—is that soot of varying maturity was accessed in a given study. As is suggested by Fig. 3 and the discussion in Section 5.1.3 (and in the response to the next comment), the MAC likely increases with maturation. Thus, in cases where literature MAC values are on the lower side, it could reflect sampling of less mature soot. Of course, differences could simply result from experimental uncertainty and historical challenges in measuring both BC concentrations and absorption accurately.

While these data support the conclusion presented in the abstract, it appears from Figure 3 that RDG theory over-predicts the MAC at smaller particle sizes when Mie Theory appears to do a better job, would a combination of these two separate approaches result in a better parameterization? It seems to be that the problem with black carbon absorption may simply be shifted from an underestimation with Mie Theory to and overestimation with RDG.

While this is certainly possible, we believe that the more likely reason for the overestimate at small sizes results from challenges with generating particles with constant composition ("maturity") at all sizes using the flame generators here. This is directly related to the discussion of soot "maturity" in section 5.1.3. While we do not have definitive proof, we believe that had we been able to sample soot of a constant maturity at all sizes we would have observed a constant (size-independent) MAC. Soot maturity has been studied for flames and fuels such as that used here by others (Johansson et al., 2017;Leschowski et al., 2015;López-Yglesias et al., 2014). The work by Johansson et al. (2017) suggests that the absorption cross-section (and thus the MAC) increases with maturity, consistent with our

observations. Additionally, they note that the wavelength dependence of absorption is steeper (higher AAE) for less mature soot. This is consistent with our observations in Fig. 4, where there is, perhaps, a slight downward trend in the AAE with particle size.

An important question is, of course, what is the maturity of soot produced from diesel engines and biomass combustion, does it vary with size, and how does it relate to the soot produced here? As we noted, "flame-generated BC particles have been shown to be a suitable proxy for atmospheric BC particles, both in terms of chemical bonding and structural properties (Slowik et al., 2007;Hopkins et al., 2007)." However, the question of size-dependent maturity for diesel and biomass burning particles is unresolved. Yon et al. (2011)investigated soot produced from combustion of diesel fuel, albeit produced from a flame generator and not an engine, but did not look at the size dependence. Their results do, however, suggest that the soot from diesel soot can be considered reasonably mature.

It's noted in the conclusions that future works will consider coatings, I look forward to seeing this work.

Indeed. We hope to have this completed soon.

**Specific comments**

Pg 2, Line 6: consider defining Mie size parameter explicitly.

We have added an explicit definition here, and on page 12. "(*The dimensionless size parameter*  $x = \pi d_p / \lambda$ , where  $d_p$  is particle diameter and l is wavelength.)"

Pg 3, Line 19: 'fuller' seems out of place, perhaps 'more complete'?

We have updated to "more thorough."

Pg 4, Line 10: what is the associated uncertainty the fuel equivalence ratio?

We have updated this as follows:

"During BC3, BC3+, and BC4, most experiments were conducted using particles produced from an inverted co-flow diffusion flame operating on methane with a sheath flow mixture of  $O_2$  and  $N_2$  (Stipe et al., 2005) with a net fuel equivalence ratio,  $\phi$ , = 0.7 ± 0.07; because the diffusion flame entrains sheath oxygen into the methane-rich center flow a range of  $\phi$  values are accessed including regions where the local  $\phi$  may be greater than 1."

Pg 8, Line 4: 'size-selected', mobility selected?

Yes. We have clarified to now state "DMA size-selected."

Pg 8, Line 4: Which soot, i.e. what lot from which company? There is a certain amount of variability?

We now report that we used Alfa Aesar Lot L18U002.

Pg 8, Line 4: Was any coating analysis performed on the data collected from the SP2?

No, we did not perform coating analysis using the SP2 for this study. The laboratory environment allowed us to constrain the amount of coating using the CPMA.

Pg 9, Line 18: perhaps citing the original paper (Park et al. 2004 I believe, there may have been one earlier that used the formulation shown in this manuscript) that brought up the Df,m mobility shape factor would be more appropriate than Cross et al. 2010.

We now cite Park et al. (2003), who report this relationship.

Pg 12, Section 5.1.1: As I understand, the methane diffusion flame was used in BC3, BC3+, and BC4. Can you explain why the Df described here is different than that of the Df described in Bhandari et al., 2016? I would expect nascent soot particles to have a lower mass mobility exponent.

First, we have updated this reference to Bhandari et al. (2017). There is fundamental difference between the fractal dimension (what Bhandari et al measure) and the mass-mobility exponent (what is measured in this study). The difference between the two is described in detail by Sorensen (2011). The fractal dimension of typical soot is on the order of ~1.8; the mass-mobility exponent of typical uncoated soot is on the order of ~2.5 (e.g. Figure 3 in Olfert et al. (2017)). Thus we fully expect the fractal dimension measured by Bhandari (2017) to be lower than the mass-mobility exponent measured here.

To the reviewer's question, the difference likely results from different methods of determination. Bhandari et al. (2017) determined the Df (fractal dimension) values from analysis of SEM images and by fitting a line to the number of spherules comprising a particle versus dimensions of the particle (specifically  $\sqrt{(L \cdot W)/d_p}$  where L and W are the length and width of the overall particle and  $d_p$  is the spherule diameter). In contrast, we determined Df,m (mass mobility exponent) from plots of per-particle mass versus mobility diameter.

We have added the following to Section 4.1: "(The mass mobility exponent,  $D_{f,m}$ , differs from the fractal dimension,  $D_f$ , as discussed by Sorensen (2011).)"

Pg 13, Line 27: It appears that Mie also overestimates at smaller particle sizes as well.

We have updated this sentence to clarify: "When the fits are restricted to x < 0.9, a reasonable fit using Mie theory is obtained at all wavelengths over this size range, *although there is perhaps a small overestimate at the smallest sizes.*"

Pg 14, Line 6: Can you comment on why these RI values are larger than those used in current global climate models?

In short, because many models have adopted the values of Bond and Bergstrom (2006), and their suggested RI yields MAC values at many sizes lower than is supported by observations. We have added the following for clarification: "For reference, using RI = 1.95 - 0.79i the *MAC* at 532 nm calculated for BC in the small particle limit (assuming a material density of 1.8 g/cm3) is only 5.1 m2 g-1, but peaks at 7.5 m2 g-1 around  $d_{p,VED} = 150$  nm."

For additional consideration, the value suggested by Bond and Bergstrom (2006) was arrived at after consideration of a variety of literature measurements of soots of various types, and with measurements made in many different ways. They determined a value of 1.95 - 0.79i by matching, approximately, a model that accounts for varying "void fractions" with a suite of the observations, and finding where this intersects with an estimate of how the refractive index varies with extent of "graphitization." They ultimately state that "The value 1.95–0.79i merely provides agreement with many of the measurements."

Pg 14, Line 8: Specify that this recommendation is for 550 nm.

It is our understanding that the RI reported by Bond and Bergstrom (2006) is not for a specific wavelength, but is meant to be general. The references used to derive the RI do not all use 550 nm. The recommended MAC value is, however, wavelength-specific.

Pg 14-15, Line various: considerable time is spent discussing the MAC values in context with other literature, but there is little comment as to why these differences may exist.

We think the most likely reason for differences is differences in soot maturity. We have added a sentence to this effect. "One key reason that differences may exist between studies is that the BC particles sampled had differing maturity. Soot maturity refers to the extent to which the BC has a more disordered internal structure with high hydrogen content (low maturity) versus a more ordered, graphite-like structure with low hydrogen content (high maturity) (Johansson et al., 2017). The absorption cross-section for BC likely increases with increasing soot maturity (López-Yglesias et al., 2014)."

Pg 17, Line 16: I suggest adding this note "RI fitting was performed. . ." to the caption in Figure 3.

We have updated Fig. 3 caption.

Pg 19, Line 1: Could information regarding the soot maturity be accessed using LEO- fitting analysis with the SP2? At least that would indicate coating thickness which would indicate atmospheric processing.

To clarify, the concept of "soot maturity" is not related to how coated (or not) a particle is, but to the properties of the BC material (see above). As such, LEO-fitting would not be especially helpful.

Figure 2: As presented the fits are hidden behind the data, please bring them to the front to allow the reader to clearly see them. Perhaps a bottom panel showing the % difference or some such metric

indicating the deviation from model vs measurement below each plot to clearly identify differences between Mie and RDG could be helpful?

We have updated Fig. 2 to move the fit lines to the front. We have not added additional % difference plots because Fig. 3 shows the transformation of the data in Fig. 2 to MAC space, and we think that this further helps to illustrate the differences.

**Response to Reviewer #2**

The manuscript presents a thorough description of optical property measurements con- ducted during a series of experiments examining soot emissions from two different types of flames. It presents a detailed analysis of the merits of Mie and RDG ap- proximations of uncoated soot optical properties, and explores implications for climate models. The experiments and measurements are of a high quality and provide an ex- tremely useful collection of data for interpretation of similar measurements performed for other BC emission sources. They raise important questions regarding treatment of BC in climate models, and I look forward to their future work related to coated BC properties

I recommend its publication in ACP with only a few minor corrections, listed below.

The reference in the introduction giving an upper end estimate of potential BC forcing is now 10 years old, and this section would benefit form including one or two more recent estimates, though I understand the authors are pointing out an extreme case.

**We have updated this as follows:**

"The exact magnitude of the climate impacts of BC remain uncertain. One estimate puts top-ofthe-atmosphere direct forcing by BC as high as 0.9 W m-2, which is comparable in magnitude to that of CO2 (Ramanathan and Carmichael, 2008). Other more recent assessments yield 0.71 W m-2 with 90% uncertainty bounds of 0.08 to 1.27 W m-2 (Bond et al., 2013) or 0.61 [+0.16 to +1.40] W m-2 (Wang et al., 2016), while the IPCC suggests a value of 0.40 [+0.05 to +0.80] W m-2 (Boucher et al., 2013)."

Page 8, line 4: please be specific as to what "size" is referring to here...mobility diam- eter or mass.

**We now state "DMA size selected" to clarify.**

Page 8, line 22: stating truncation angles for the CAPS-SSA here would be helpful.

Rather than giving the truncation angles, we now state the magnitude of the correction based on (Onasch et al., 2015b).

"The CAPS  $PM_{SSA}$  measures  $b_{sca}$  using an integrating nephelometer, corrected for the finite viewing angle of the detector, i.e. truncation correction (Onasch et al., 2015b). *The truncation*

correction at 630 nm was determined to be <1% at 630 nm for particles smaller than 300 nm, increasing to <5% for particles smaller than 800 nm."

**Page 9 - i believe the equations for MAC should have units of area, not inverse Mm**

We think the reviewer meant cross-sections, not MAC. We have updated the manuscript.

**Response to Reviewer #3**

This is a well written article and present the solid absorption result using the well con- strained soot source and the result is straightforward for models to pick up. but it would even more benefit the community if addressing the following points:

-how could flame soot represent the ambient soot, in terms of refractive index and particle morphology? Then how could be suggested these results could be widely used in the model?

-if we have a different source of BC, for example the biomass burning, how could we guarantee the RI still the same?

We address the above two points together. While we do not have direct evidence that the BC produced from our flames is guaranteed to have the same RI as ambient soot, we point to the references of Hopkins et al. (2007) and Slowik et al. (2007) as support for the lab soot and ambient soot having similar properties. As we previously stated:

"Although atmospheric BC particles are predominately generated through combustion of fossil fuels or through biomass burning (Bond et al., 2013), flame-generated BC particles have been shown to be a suitable proxy for atmospheric BC particles, both in terms of chemical bonding and structural properties (Slowik et al., 2007;Hopkins et al., 2007)."

We have expanded this discussion (in Section 5.1.3) to include additional details:

"For example, Hopkins et al. (2007) find that the sp2 content of ethylene and methane flame soot are similar to diesel soot (63%, 60%, and 56%, respectively), and have similar aromatic content. There is also a reasonable similarity between SP-AMS mass spectra of flame soot and soot particles in diesel exhaust or smoke from biomass burning (Onasch et al., 2015a)."

-It would be better to show the mass distribution of DMA-selected particles at different cases (to indicate the width of the distribution), as the single particle mass is crucial for the following analysis.

We now report typical geometric standard deviations ( $\sigma_{g,CPMA}$ ) in the measured per-particle mass from the CPMA. For "forward-coating" experiments, the typical  $\sigma_{g,CPMA}$  was 1.3. For "reverse-coating" experiments, the  $\sigma_{g,CPMA}$  were generally larger, ranging from around 1.3 to 2. Based on the reviewers concern, we have examined whether there is any dependence of the measured MAC values on the

 $\sigma_{g,CPMA}$ . Using data from BC4 as an example, we find no significant dependence of the measured MAC values on  $\sigma_{g,CPMA}$ , with a linear fit between MAC and  $\sigma_{g,CPMA}$  giving an  $r^2

-there is no label for Fig. S8.

Perhaps there was a pdf rendering problem? We downloaded the supplement and can see the figure caption for Fig. S8.

-one important information is how the size parameter could relate to the volume equivalent diameter. For a general practice, could we assume >160nm BC will have a MAC using RDG approach, and how this VED will depend on the wavelength.

The size parameter relates to the volume equivalent diameter as:  $x = \pi d_{p,VED}/\lambda$ . We now give this definition explicitly. Yes, for larger particles it can be assumed that the MAC of BC can be represented using an RDG approach. This is equivalent to assuming a constant MAC, as we conclude in the last sentence of our abstract.

**Measurement and modeling of the multi-wavelength optical properties of uncoated flame-generated soot**

Sara D. Forestieri,1, # Taylor M. Helgestad1,# Andrew Lambe,2,3 Lindsay Renbaum-Wolff2, Daniel A. Lack,4,5,^ Paola Massoli,2 Eben S. Cross,6,& Manvendra K. Dubey,7 Claudio Mazzoleni,8 Jason

5 Olfert,9 Arthur Sedlacek,10 Andrew Freedman,2 Paul Davidovits,3 Timothy B. Onasch,2,3 Christopher D. Cappa1

1Department of Civil and Environmental Engineering, University of California, Davis, CA 95616 2Aerodyne Research Inc., Billerica, Massachusetts, USA, 01821 3Chemistry Department, Boston College, Boston, MA, USA, 02467

[revised manuscript text omitted]
 *MACs* at  $\lambda = 532$  nm of 6.7  $\pm$  0.7 m2g-1 for  $d_m = 155$  nm particles and 8.7  $\pm$  0.1 m2g-1 for  $d_{\rm m} = 320$  nm particles. This general behavior was also observed for soot particles generated

| sed        |                           |
|------------|---------------------------|
| oth        |                           |
| in         |                           |
| tly_       | Deleted: ,                |
| 5 – |                           |
| ity        | Formatted: Font: Italic   |
|            |                           |
|            |                           |
|            |                           |
| ice        |                           |
| cle        |                           |
| ted        | Deleted: Figure 2         |
| r a        | Formatted: Font: Not Bold |
| ter        |                           |
| ies        | Formatted: Font: Not Bold |
| m,         | Deleted: Figure 3         |

from a methane diffusion flame in Dastanpour et al. (2017), with *MACs* values reported at  $\lambda = 660$  nm of ~5 m2 g-1 for  $d_{p,VED} = 50$  nm and ~7 m2 
[revised manuscript text omitted]

|   |        |           |                    |           |                        |                  | 1410                                | dp VED, peak        | No. of         |
|---|--------|-----------|--------------------|-----------|------------------------|------------------|-------------------------------------|---------------------|----------------|
|   | Study# | Soot Type | Meth
od         | λ
(nm) | I n strument    | n = m + ki       | $MAC_{peak}$
(m 2 /g) | Param. @ | data
points |
| - | BC2    | Nascent   | Mie%               | 405       | PASS-3                 | 2.21 + 0.86i     | 10.45 (±2.06)                       | 123/0.95            | 36             |
|   | BC2    | Denuded   | Mie                | 405       | PASS-3                 | 2.19 + 0.91i     | 10.68 (±1.97)                       | 119/0.92            | 30             |
|   | BC2    | Nascent   | RDG \$, | 405       | PASS-3                 | 1.80 + 1.13i     | 10.06 (±2.22)                       |                     | 36             |
|   | BC2    | Denuded   | RDG                | 405       | PASS-3                 | 1.80 + 1.18i     | 10.32 (±2.56)                       |                     | 30             |
|   | BC2    | Nascent   | Mie                | 532       | PASS-3
NOAA CRD-    | 2.13 + 0.64i     | 6.92 (±0.78)                        | 178/1.07            | 36             |
|   | BC2    | Nascent   | Mie                | 532       | PAS                    | $2.39 \pm 0.79i$ | 8.02 (±0.13)                        | 169/0.99            | 43             |
|   | BC2    | Denuded   | Mie                | 532       | PASS3
NOAA CRD-     | $1.96 \pm 0.83i$ | 7.70 (±0.90)                        | 150/0.89            | 31             |
|   | BC2    | Denuded   | Mie                | 532       | PAS                    | 2.56 + 1.11i     | 9.00 (±0.80)                        | 152/0.90            | 46             |
|   | BC2    | Nascent   | RDG                | 532       | PASS-3                 | 1.80 + 0.78i     | 6.92 (±0.43)                        |                     | 36             |
|   |        |           |                    |           | NOAA CRD-              |                  |                                     |                     |                |
|   | BC2    | Nascent   | RDG                | 532       | PAS                    | $1.80 \pm 0.85i$ | 6.16 (±1.75)                        |                     | 43             |
|   | BC2    | Denuded   | RDG                | 532       | PASS-3                 | 1.80 + 1.08i     | 7.35 (±1.69)                        |                     | 31             |
|   | BC2    | Denuded   | RDG                | 532       | CRD-PAS                | 1.80 + 1.13i     | 7.55 (±0.68)                        |                     | 46             |
|   | BC2    | Nascent   | Mie                | 781       | PASS-3                 | 2.16 + 0.76i     | 5.10 (±0.40)                        | 250/0.96            | 31             |
|   | BC2    | Denuded   | Mie                | 781       | PASS-3                 | 2.84 + 0.74i     | 6.20 (±0.20)                        | 239/0.95            | 36             |
|   | BC2    | Nascent   | RDG                | 781       | PASS-3                 | $1.80 \pm 0.50i$ | 2.59 (±0.33)                        |                     | 36             |
|   | BC2    | Denuded   | RDG                | 781       | PASS-3                 | 1.80 + 0.73i     | 3.64 (±0.81)                        |                     | 31             |
|   | BC3+   | Denuded   | Mie                | 405       | UCD CRD-
PAS        | 2.11 + 1.03i     | 11.09 (±2.82)                       | 110/0.85            | 27             |
|   | BC3+   | Denuded   | RDG                | 405       | UCD CRD-
PAS        | 1.80 + 2.05i     | 10.78 (±3.11)                       |                     | 27             |
|   | BC3+   | Denuded   | Mie                | 532       | UCD CRD-
PAS        | 1.46 + 0.54i     | 6.03 (±1.12)                        | 118/0.70            | 22             |
|   | BC3+   | Denuded   | RDG                | 532       | UCD CRD-
PAS        | 1.80 + 0.76i     | 5.61 (±0.46)                        |                     | 22             |
|   | BC3+   | Denuded   | Mie                | 630       | CAPS PM SSA | 1.78+0.45i       | 3.93 (±0.90)                        | 188/0.93            | 22             |
|   | BC3+   | Denuded   | RDG                | 630       | CAPS PMssa             | $1.80 \pm 0.55i$ | 3.51 (±1.55)                        |                     | 22             |

**Table 2.** Theory-specific effective refractive indices for ethylene premixed flame soot from BC2 and BC3+ retrieved via fitting Mie Theory and the RDG approximation to the  $\sigma_{abs}$  observations. Nascent and denuded experiments are considered separately.

% Uncertainties in *MACs* are  $1\sigma$  from the least  $\chi^2$  fit. See Supplementary Figures S7-8.

 $^{s}$  The *n* values from the RDG method are non-unique. Therefore, uncertainty estimates from this work are not available. See text for details.

(a) VED and the size parameter ( $x = \pi d_p / \lambda$ ) where the peak *MAC* occurs

\* There are many degenerate RI combinations that give similar quality fit to RDG theory. Thus, a value of 1.80 was chosen for the effective real refractive index.

**In BC2 the ethylene flat-burner flame was sampled 20.3 cm between the burner surface and the sampling inlet and during BC3 the flame was sampled 2" between the burner surface and the sampling inlet.**